# IL1R1$^+$ cancer-associated fibroblasts drive tumor development and immunosuppression in colorectal cancer

E. Koncina [1,9], M. Nurmik[1,9], V. I. Pozdeev[1,9], C. Gilson [1], M. Tsenkova[1], R. Begaj[1], S. Stang[2], A. Gaigneaux [1], C. Weindorfer[2], F. Rodriguez[1], M. Schmoetten[1], E. Klein[1], J. Karta[1], V. S. Atanasova[2], K. Grzyb [3], P. Ullmann[1], R. Halder [3], M. Hengstschläger[2], J. Graas[4], V. Augendre[5], Y. E. Karapetyan[6], L. Kerger[7], N. Zuegel[7], A. Skupin [3], S. Haan [1], J. Meiser [8], H. Dolznig [2]✉ & E. Letellier [1]✉

Fibroblasts have a considerable functional and molecular heterogeneity and can play various roles in the tumor microenvironment. Here we identify a pro-tumorigenic IL1R1$^+$, IL-1-high-signaling subtype of fibroblasts, using multiple colorectal cancer (CRC) patient single cell sequencing datasets. This subtype of fibroblasts is linked to T cell and macrophage suppression and leads to increased cancer cell growth in 3D co-culture assays. Furthermore, both a fibroblast-specific *IL1R1* knockout and IL-1 receptor antagonist Anakinra administration reduce tumor growth in vivo. This is accompanied by reduced intratumoral Th17 cell infiltration. Accordingly, CRC patients who present with *IL1R1*-expressing cancer-associated-fibroblasts (CAFs), also display elevated levels of immune exhaustion markers, as well as an increased Th17 score and an overall worse survival. Altogether, this study underlines the therapeutic value of targeting IL1R1-expressing CAFs in the context of CRC.

Over the last decade, our understanding of the role of the tumor microenvironment (TME) in cancer development has significantly increased. It has now been established that the various environmental aspects of the TME, such as hypoxia, acidity, and extracellular matrix (ECM) composition, as well as the various cells present, such as macrophages and dendritic cells, play a significant role in tumor development[1]. One of the key players in the TME, which have risen to particular prominence are cancer-associated fibroblasts (CAFs). CAFs have been linked to almost all major hallmarks of cancer[2] and their presence is a significant predictor of tumor recurrence and progression in several different tumor types, including colorectal cancer (CRC)[3].

Even as the study of their functional role in tumor development and progression has increased in importance, the molecular characterization of CAFs and their identification has lagged. We and others have highlighted the difficulties associated with the usage of typical CAF/fibroblast markers due to their heterogeneous expression within the general CAF population and their non-specificity in regard to different cell types within the TME[4,5]. In addition, recent studies have emphasized that the CAF population

[1]Molecular Disease Mechanisms Group, Department of Life Sciences and Medicine, University of Luxembourg, Belval, Luxembourg. [2]Center for Pathobiochemistry and Genetics, Institute of Medical Genetics, Medical University of Vienna, Vienna, Austria. [3]Luxembourg Centre for Systems Biomedicine, University of Luxembourg, Belval, Luxembourg. [4]Clinical and Epidemiological Investigation Center, Department of Population Health, Luxembourg Institute of Health, Luxembourg, Luxembourg. [5]National Center of Pathology, Laboratoire National de Santé, Dudelange, Luxembourg. [6]Integrated BioBank of Luxembourg, Dudelange, Luxembourg. [7]Department of Surgery, Centre Hospitalier Emile Mayrisch, Esch-sur-Alzette, Luxembourg. [8]Cancer Metabolism Group, Department of Cancer Research, Luxembourg Institute of Health, Luxembourg, Luxembourg. [9]These authors contributed equally: E. Koncina, M. Nurmik, V. I. Pozdeev. ✉e-mail: helmut.dolznig@meduniwien.ac.at; elisabeth.letellier@uni.lu

can be further subdivided into subgroups with differing functional activities[6–9]. The gene expression of various markers, such as fibroblast activation protein (FAP)[8] and podoplanin (PDPN)[9], is extremely variable between CAF subtypes in different tumor types, further complicating the development of a single versatile solution for the study of the tumor stromal compartment. In PDAC for instance, two main CAF subtypes have been described. While a first subtype highly expresses α-smooth muscle actin (αSMA), a second subtype is characterized by low αSMA expression and high expression of IL−6. Both subtypes were respectively named myCAFs and iCAFs by the authors to reflect the high expression of myofibroblastic markers such as αSMA by the former and the pronounced inflammatory nature of the latter[6].

To dissect the high heterogeneity and functional variance present in the fibroblast population, studies have turned towards single-cell sequencing, which has become a key technique in CAF characterization. For example, single-cell sequencing of breast cancer tissues identified a subpopulation of CAFs, entitled CAF-S1, which showed immunosuppressive properties by attracting T-lymphocytes, promoting their differentiation into T regulatory cells (Tregs), and by further boosting the capacity of Tregs to inhibit effector T cells[7]. In a follow-up study, this CAF-S1 population was further subdivided into eight subtypes, all with their own unique gene expression profiles and predicted functionality[10]. Another very recent study has identified LRRC15+ fibroblasts to have immunosuppressive effects in pancreatic cancer[11]. While this demonstrates the incredible diversity of CAFs, it also highlights the importance of characterizing their functional roles in tumor development.

One of the pathways involved in the development and pro-tumorigenic nature of CAFs is the IL-1 pathway. The receptor IL1R1 binds IL-1α, IL-1β, and IL1RN. Once IL-1α or IL-1β is bound to IL1R1, it forms a heterotrimeric complex with the co-receptor IL1RAP and induces the downstream signaling cascade. This signaling pathway has been shown to shape CAF heterogeneity and to drive the iCAF phenotype[12]. Additionally, the IL−1 pathway has been linked to tumorigenicity, including chemotherapy resistance[13,14]. It was demonstrated that IL-1α is involved in chemoradiotherapy resistance in rectal cancer by inducing p53-mediated therapy-induced senescence of iCAFs[14], which recently led to the initiation of a clinical trial[15]. Nevertheless, the activation and functions of the IL-1 signaling pathway in CAFs remains to be determined.

In this study, we identify a CAF subtype in CRC, characterized by high IL1R1 expression and elevated IL−1β-driven signaling in various single-cell sequencing datasets. We show that, in the TME, IL1R1 expression is strongly and specifically linked to CAFs and correlates with the expression of signature genes, such as FAP and PDPN. Elevated levels of IL1R1 are associated with alterations in immune cell signatures and exhaustion markers, suggesting that IL1R1+ fibroblasts may play an immunomodulatory role in the TME. Accordingly, we show that IL-1β-stimulated fibroblasts induce the polarization of macrophages, most probably via the secretion of monocyte chemoattractant protein-1 (MCP-1/CCL2), towards a pro-tumorigenic (M2-like) phenotype and reduce T cell proliferation. In addition, we demonstrate that IL-1β stimulated fibroblasts induce tumor growth in 3D organotypic tumor assays, an effect which is abrogated upon IL−1 signaling blockade. The fibroblast-specific ablation of IL-1 signaling in vivo results in higher survival and reduced tumor growth, accompanied by a lower infiltration of pro-tumorigenic Th17 cells and an overall reduced immunosuppressive TME. Finally, the identified IL1R1+ CAF subtype is correlated with lowered survival in CRC patients belonging to Consensus Molecular Subtype 4 (CMS4)−a molecular subtype of CRC that shows the lowest therapy response and overall survival.

## Results

### Single-cell sequencing analysis identifies IL1R1 expression and the IL-1 pathway as a key player in colonic CAFs

To investigate the role of the IL-1 pathway in colonic CAFs, we created a single-cell sequencing library of our primary CRC patient samples (Cole) and integrated it with two previously published datasets (Li[8] and Zhang[16]) to generate the CLZ dataset. In addition, we analyzed two more recently published CRC single-cell datasets (Lee[17] and Qian[18]). We clustered all three datasets and assigned the main cell type identities (Fig. 1a and Supplementary Fig. 1a for the number of cells analysed and Supplementary Data 1 for the associated metadata). We first analyzed the expression of the IL-1 pathway members (Supplementary Fig. 1b) and observed that fibroblasts are the main IL1R1-expressing cell type (Fig. 1b), suggesting that they might represent the most IL-1-signaling-sensitive and responsive cells in CRC. High IL1R1 mRNA expression in the tumor stroma compared to the epithelial compartment was also detectable in three additional public bulk CRC datasets, where stromal and epithelial cells were either isolated by fluorescence-activated cell sorting (FACS), based on FAP and EPCAM expression (Calon[19]), or by laser microdissection (Nishida[20] and Rupp[21,22]) (Supplementary Fig. 1c). In addition, we observed that the expression of IL1B, one of the activating ligands of the IL-1 pathway (Supplementary Fig. 1b), was significantly higher in fibroblasts in the Rupp and Nishida datasets (Supplementary Fig. 1c). Of note, we could not detect any consistent difference in the expression of the other members of the IL-1 pathway, including IL1A, which has previously been associated with iCAFs in both PDAC and chemoradiotherapy resistant rectal cancer[12,14] (Supplementary Fig. 1c). Using patient-derived CAF and tumor spheroid cultures, we confirmed that CAFs expressed more IL1R1 than tumor cells (Supplementary Fig. 1d and Supplementary Table 1). We found a strong correlation between IL1R1 expression and a stromal signature score (ESTIMATE)[23] (Supplementary Fig. 1e) in the Cancer Genome Atlas (TCGA) dataset. Additionally, we found that CMS4 patients, characterized by tumors with high stroma infiltration and poor prognosis, expressed significantly higher levels of IL1R1 than CMS1, 2 and 3 patients (Supplementary Fig. 1f). Next, we assessed the expression levels of IL1B and both members of the IL-1 receptor, IL1R1 and IL1RAP in normal fibroblasts (NFs) and CAFs (Fig. 1c lower panels) and observed a significantly higher expression in CAFs compared to NFs (Fig. 1C). We did not observe such a difference for IL1A, nor for the two inhibiting molecules of the pathway−IL1RN and IL1R2 (Supplementary Fig. 1g). Using primary cultures of NFs and CAFs, we further demonstrated increased cell-surface expression of IL1R1 protein in CAFs when compared to NFs (Fig. 1d), although to varying degrees depending on the patient (Supplementary Fig. 1h).

We then wondered whether the higher expression of the receptor was associated with an increased IL-1 signaling in CAFs. The differential gene expression (DGE) analyses comparing IL-1β-stimulated fibroblasts to their non-stimulated counterparts showed that CAFs were more sensitive to IL-1β than NFs (Fig. 1e). We then selected the top fifty genes upregulated in IL-1β-stimulated fibroblasts to establish an IL-1β score (Supplementary Table 2). We found the IL-1β scores to be significantly higher in CAFs when compared to NFs, suggesting that increased IL-1 signaling is also occurring in vivo in CAFs within the tumor tissues (Fig. 1f). Similarly, we confirmed NFκB target genes to be upregulated in CAFs compared to NFs (Supplementary Fig. 1i).

### Members of the IL-1 pathway are strongly associated with IL1R1+ iCAFs and lower survival in CMS4 patients

To identify CAF subpopulations with a specific IL-1 pathway member expression profile, we further subclustered tumor fibroblasts based on their gene expression profiles (Supplementary Fig. 2a). Clusters were grouped into myofibroblastic CAFs, or myCAFs (expressing ACTA2, RGS5, CSPG4, and AOC3) and inflammatory CAFs, or iCAFs

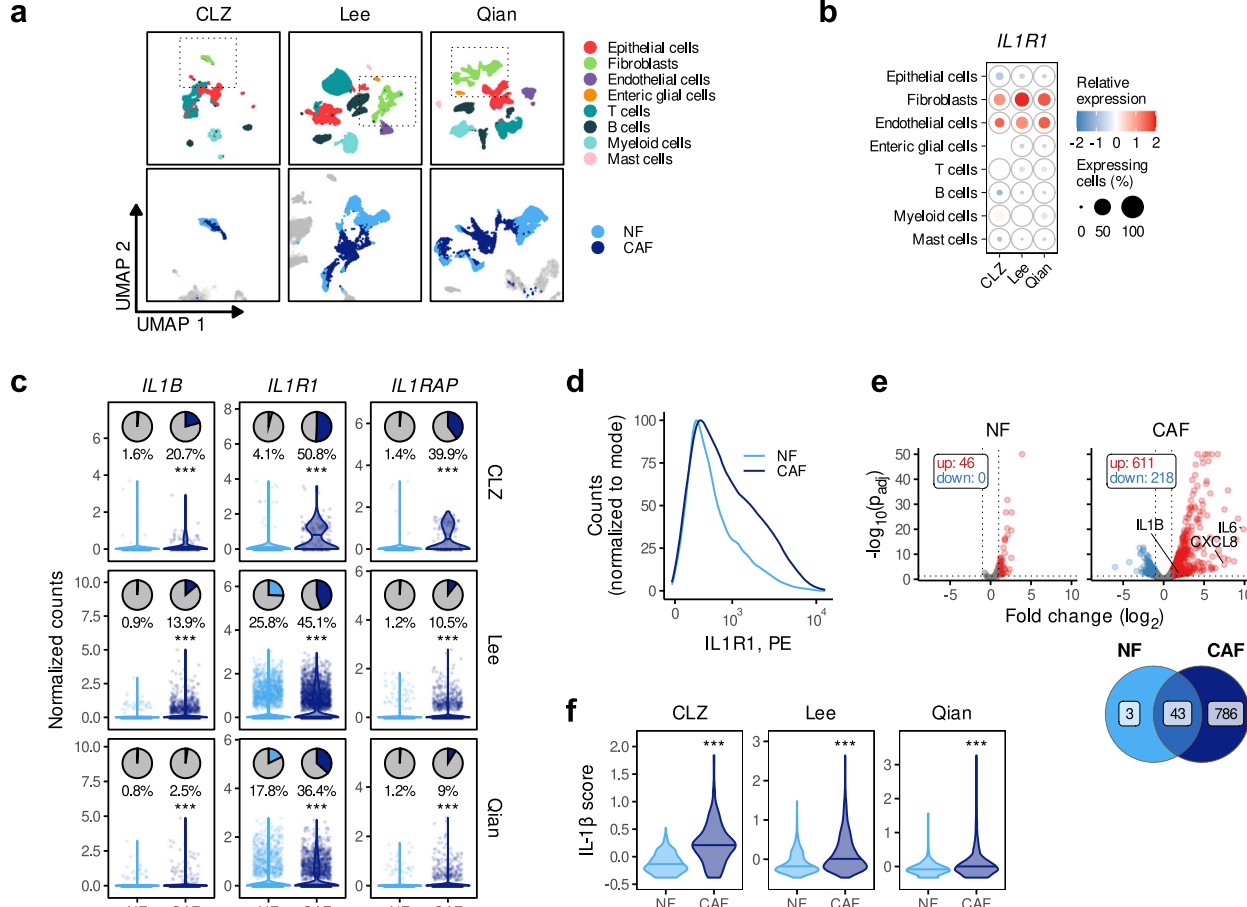

**Fig. 1 | *IL1R1* and *IL1B* expression in CRC patients. a** UMAP plot showing the main cell types identified in the CLZ, Lee, and Qian scRNA-seq datasets (upper row) and the distribution between normal and tumor fibroblasts (lower row). **b** Expression of *IL1R1* across the different cell-types in the three scRNA-seq datasets. **c** Expression of *IL1R1*, *IL1RAP*, and *IL1B* in the three scRNA-seq datasets. The pie charts show the percentage of *IL1R1-*, *IL1RAP-* or *IL1B*-expressing cells (counts > 0) and the violin and scatter plots show the normalized counts in the NF and CAF populations. **d** Expression of IL1R1 in paired primary normal and tumor fibroblasts from a CRC patient (P42, patient characteristics in Supplementary Table 1) determined by flow cytometry. **e** *Upper part*—Volcano plot showing the genes differentially expressed upon IL-1β treatment in paired NFs and CAFs from *n* = 2 CRC patients (P20 and P42). *Lower part*—Venn diagram showing the overlap of the number of differentially expressed genes in NFs and CAFs upon IL-1β treatment (1 ng/ml). 46 and 829 genes were differentially expressed in NFs and CAFs respectively, among which 43 genes were upregulated in both cell types. **f** IL-1β scores in normal and tumor fibroblasts from the three datasets. Statistical differences were calculated using the two-sided Wilcoxon signed rank test in **c** and **f** (\*\*\**p* < 0.001). Horizontal lines in C and F show the median. Number of patients per dataset in panels a-c and f is reported in Supplementary Fig. 1a. Source data are provided as a Source Data file.

(expressing cytokines such as *CXCL14, CXCL12, CXCL16*) (Supplementary Fig. 2b) in agreement with previous studies. In line with a known limitation of scRNA-sequencing, we were not able to discriminate myCAFs and pericytes as both populations express common gene markers, in particular *RGS5*, also attributed to pericytes[24,25]. We then looked for *IL1R1* and found these cells to be predominant in the iCAF group and distinguished IL1R1+ iCAFs (*IL1R1*high population) from the remainder iCAFs (*IL1R1*low population) (Supplementary Fig. 2b). We were able to consistently identify these three subgroups in all three scRNA-seq datasets (Fig. 2a, b). In addition, when integrating all three datasets, we confirmed that the identified *IL1R1*-expressing fibroblast subtype was a common cluster shared among the different cohorts (Cluster 7 in Supplementary Fig. 2c–e). Moreover, IL1R1+ iCAFs are characterized by the highest IL-1β scores in all datasets, indicating that in addition to expressing more *IL1R1*, IL1R1+ iCAFs are characterized by induced IL-1-driven signaling, most likely due to increased receptor availability (Fig. 2c). Finally, IL1R1+ iCAFs do not only express common CAF, and in particular iCAF, markers, such as *FAP* and *PDPN*, but also ECM-modulating molecules like *MMP2*[26], as well as increased amounts

of tumorigenic secretory molecules, such as *CXCL12*[27] and *IGF1*[28] (Fig. 2d).

DGE analyses were used to establish iCAF and IL1R1+ iCAFs gene signature lists (Supplementary Table 2). Of note, we deliberately excluded *IL1R1* from the IL1R1+ iCAF signature to enable further counter validations. Both gene sets were used to calculate iCAF and IL1R1+ iCAF scores and to substantiate their specificity to stromal cells (Supplementary Fig. 3a). We then used the IL-1β and IL1R1+ iCAF gene sets to calculate both scores in the TCGA dataset and found them to be correlated, suggesting that the presence of IL1R1+ iCAF cells in tumors is linked to elevated IL-1β signaling (Supplementary Fig. 3b). *IL1R1* expression also strongly correlated with the IL1R1+ iCAF signature in CMS4 patients, while its correlation with the iCAF signature was weaker (Fig. 2e), further supporting the specificity of our IL1R1+ iCAF signature. We additionally confirmed that CAFs expressing higher levels of *IL1R1* exhibit an enrichment in inflammation-related gene sets (Supplementary Fig. 3c).

Next, we investigated the potential link between the identified CAF subpopulations and patient survival. We found that CMS4 patients in the top quartile of *IL1R1* expression had a lower overall survival rate

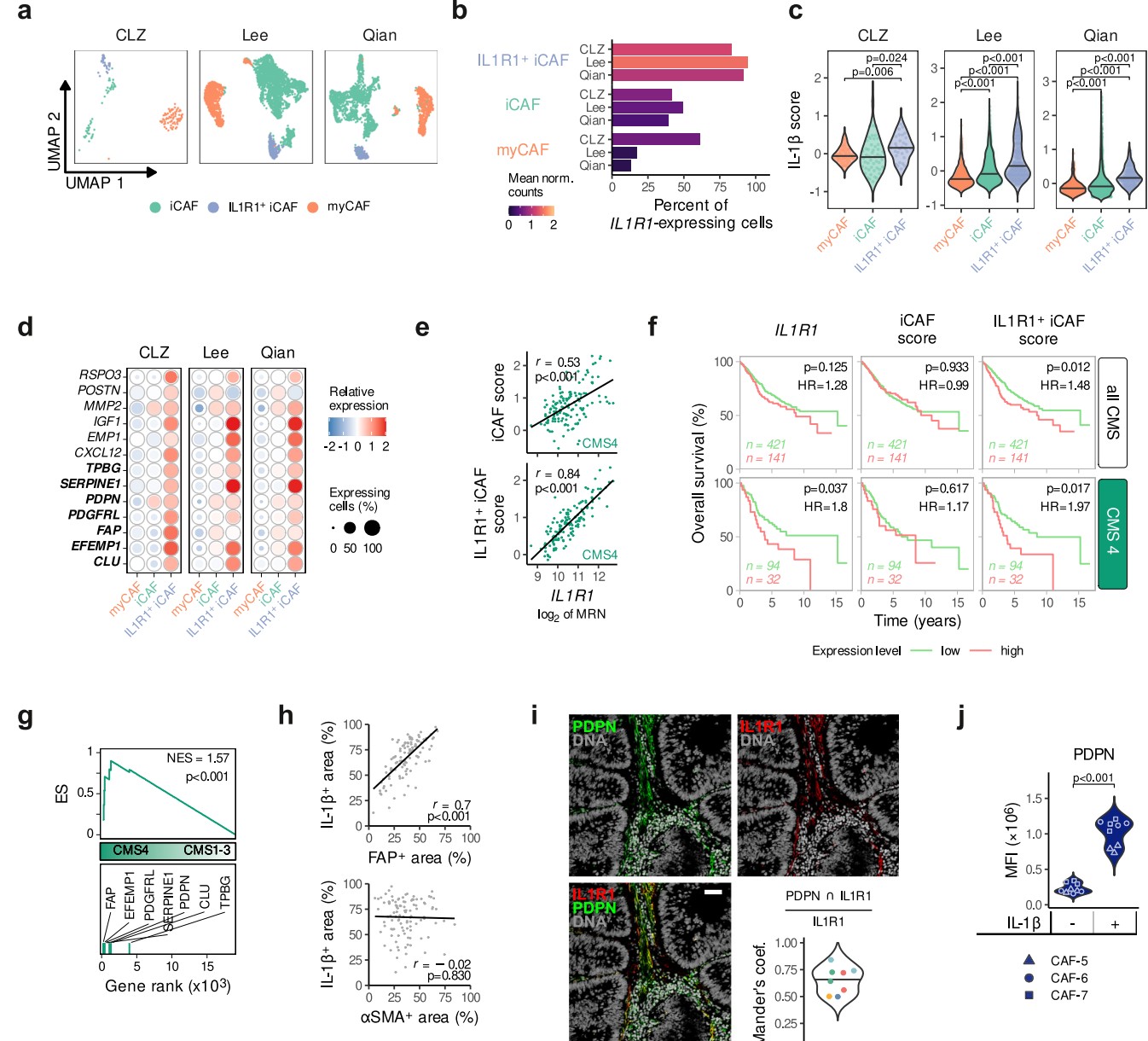

**Fig. 2 | *IL1R1* and *IL1B* expression in CAF subtypes and their association with survival in CRC patients. a** UMAP plot showing CAF subtypes identified in the CLZ, Lee and Qian scRNA-seq datasets. **b** *IL1R1* expression in the three CRC CAF subtypes (iCAF, IL1R1+ iCAF and myCAF). The bar length shows the percentage of *IL1R1*-expressing cells and the color gradient shows the mean of normalized counts. **c** IL-1β scores in the CAF subpopulations from the CLZ, Lee, and Qian datasets. Horizontal lines show the median. Holm's adjusted pairwise two-sided Wilcoxon signed rank test (***p < 0.001). **d** Heatmap showing the expression of IL1R1+ iCAF-related genes in the three CAF subtypes of the three datasets. The color gradient shows the relative expression (mean of scaled expression values) and the bubble size shows the percentage of expressing cells. Genes appearing in bold define the IL1R1+ iCAF signature. **e** Correlation between *IL1R1* expression with iCAF and IL1R1+ iCAF scores in CMS4 patients (TCGA dataset, *n* = 143). **f** Kaplan-Meier curves showing the prognostic value of *IL1R1*, iCAF and IL1R1+ iCAF scores on the overall survival of CRC patients (GSE39582 dataset). The upper row shows all patients (*n* = 562) and the lower row shows the subset of CMS4 patients (*n* = 126). The third quartile expression value was used as a threshold to split patients into low and high expressors (75% low vs 25% high expressing patients). Non-adjusted *p*-values and hazard ratio

(HR) of the Cox proportional hazard model are reported. **g** GSEA of the IL1R1+ iCAF gene set in CMS4 (*n* = 143) vs. CMS1-3 (*n* = 368) TCGA patients. The running enrichment score (ES), as well as the normalized enrichment score (NES) and *p*-value are reported. **h** Correlation between FAP+ and IL-1β+ staining (top panel) and αSMA+ and IL-1β+ staining (lower panel) identified after IHC staining on tissue microarray sections of our established in-house CRC cohort (*n* = 106 patients). **i** IL1R1 and PDPN co-localization in immunofluorescence stainings of human tumor samples (*n* = 5) and beeswarm plot showing the normalized Mander's colocalization coefficient of IL1R1 and PDPN on sections measured from five different patients (shown by different colors on the lower right panel). Individual microphotographs of IL1R1 and PDPN IHC stainings as well as DAPI stained DNA content. Scale bar = 50 μm. **j** Expression of PDPN determined by flow cytometry (MFI) in IL-1β stimulated (1 ng/ml) CAFs (CAF-5, CAF-6 and CAF-7 cultures, patient characteristics in Supplementary Table 1) and unstimulated controls. Tukey post-hoc test following a nested ANOVA design (***p < 0.001). For **e** and **h**, the Pearson's correlation coefficients *r* and two-sided *p*-values are reported. Number of patients per dataset in **a**–**d** is reported in Supplementary Fig. 1a. Source data are provided as a Source Data file.

(Fig. 2f). We observed a similar reduction in survival in CMS4 patients with high IL1R1[+] iCAF scores, while the iCAF score was not prognostic of the outcome. This indicates that the IL1R1[+] subpopulation of CAFs may possess elevated pro-tumorigenic functions when compared to other CAF subtypes (Fig. 2f). Additionally, we found that IL1R1[+] iCAF genes were enriched in CMS4 patients (Fig. 2g). IHC staining of an in-house CRC patient tissue microarray (TMA; $n = 106$, Supplementary Table 3) revealed a strong correlation of IL-1β with FAP expression (Fig. 2h), especially in CMS4 (Supplementary Fig. 3d), but not between IL-1β and αSMA expression (Fig. 2h, Supplementary Fig. 3d), further reinforcing the link between FAP-expressing CAFs and IL-1β. In addition, immunofluorescence staining of tumor histology samples from CRC patients revealed colocalization of PDPN, whose gene is part of the IL1R1[+] iCAF set, and IL1R1 in the stromal tissue (Fig. 2i, left panel for images, lower-right panel for quantification). Finally, we could confirm that stimulation with IL-1β increased PDPN expression on CAFs derived from three different CRC patients (Fig. 2j).

## An autocrine IL-1β-driven signaling loop maintains an activated state in fibroblasts

We further used a transwell assay (Supplementary Fig. 4a) to examine whether an IL-1-driven crosstalk takes place between CAFs and tumor cells. In this experimental setup, tumor spheroids and fibroblasts were co-cultured by excluding direct cell-to-cell contact, while allowing the exchange of secreted factors between both compartments. IL-1β expression at the transcript level (Fig. 3a) and protein secretion (Fig. 3b) increased in NFs upon co-culture with patient-derived 3D tumor spheroids. This effect was even more pronounced in CAFs (Fig. 3a,b), indicating that CAFs are more sensitive to IL-1 pathway activation by tumor cells than NFs. Next, the expression profiles of CAFs with high or low levels of *IL1R1* in response to the presence of tumor cells were determined. CAFs isolated from six different patients were co-cultured with tumor spheroids followed by RNA sequencing. The six CAF cultures were divided into *IL1R1*[lo] ($n = 3$ independent biological replicates from three different patients P16, P19, P22) and *IL1R1*[hi] ($n = 3$ independent biological replicates from three different patients P32, P41, P42) groups based on the median expression value of the receptor (Supplementary Fig. 4b). A DGE analysis revealed that *IL1R1*[hi] CAFs were responsive to the presence of tumors, while *IL1R1*[lo] CAFs reacted to a lower extent (Supplementary Fig. 4c). Interestingly, upregulated genes included ECM-modulating molecules similarly to the IL1R1[+] iCAF signature described before (Fig. 2d). Along the same line, IL-1β-stimulated CAFs showed enrichment in pathways related to increased growth, invasion, survival, angiogenesis and modulation of immune cells (Fig. 3c and Supplementary Fig. 4d). Additionally, we observed increased gene expression levels of *IL1B*, *IL6* and *CXCL8*/*IL8*, the latter two being known for their pro-tumorigenic roles (Fig. 3d), along with activation of the NFκB pathway (Supplementary Fig. 4e). A comparative cytokine profiling of media conditioned by untreated and IL-1β-treated CAFs showed an increased release of several pro-inflammatory cytokines, such as IL-6, CXCL8/IL-8, CXCL1, CXCL5, and CSF2/GM-CSF (Fig. 3e, f).

To further investigate the potential IL-1-NFκB axis, we analyzed the nuclear translocation of p65, a proxy for the activation of the NFκB pathway, in collagen gel co-cultures with tumor cell spheroids. Determination of the nuclear/cytoplasmic ratio (N/C) in CAF monocultures revealed increased nuclear p65 localization after treatment with IL-1β, confirming its ability to activate the NFκB pathway in CAFs (Fig. 3g). We observed a significant increase in NFκB pathway activation in CAFs upon coculture and noted that CAFs located in close vicinity to tumor cells showed the highest p65 N/C (red data points in Fig. 3g, CAF outlines in Supplementary Fig. 4f). This observation was further supported by the increased expression of known NFκB downstream targets transcripts in the RNA-Seq coculture dataset (Supplementary Fig. 4g). Interestingly, in CAFs with already a basal activation

of the NFκB pathway, the addition of the IL1R1 antagonist Anakinra to untreated CAFs reduced the nuclear presence of p65 and thereby the activation of the NFκB pathway, supporting the existence of an IL-1-driven autocrine loop in CAFs, which potentially helps maintain the pro-tumorigenicity of these cells (Supplementary Fig. 4h). Altogether, our results suggest that targeting basal IL-1β crosstalk between CAFs and colorectal tumor cells may inhibit the pro-tumorigenic function of CAFs. Along this line, IL-1β was able to promote an iCAF phenotype as shown by the increase in PDPN, PDGFRα and FAP protein expression, which was reversed upon Anakinra treatment (Fig. 3h). Activation of the TGF-β pathway is a hallmark of CMS4 tumors and there is a close interaction between the IL-1β and TGF-β signaling pathways. In agreement with the literature, we observed that TGF-β induces a myofibroblastic phenotype (expression of αSMA), which could be partially reverted by the induction of the IL-1β signaling pathway (Supplementary Fig. 4i). Interestingly, we observed that neither IL-1β nor TGF-β alone could induce the expression of FAP. However, when we simultaneously triggered both signaling pathways, we observed a large increase of FAP expression (Supplementary Fig. 4i). This finding is quite intriguing and highlights the complexity of the interaction between both pathways, which cannot be reduced to a simple negative feedback interaction alone. Moreover, there is increasing evidence for the simultaneous presence of different types of CAFs at spatially distinct sites in the TME[29], which—in line with the different subsets of CAFs identified in the scRNA-Seq datasets—strongly suggests that there are topological distinct areas that are exposed to different signals and consequently associated to different CAF phenotypes. Additional studies are required to elucidate the reasons behind the presence of diverse CAF subsets in CMS4 tumors, and more importantly, to determine their specific functions in CRC.

## IL-1β triggers pro-tumorigenic signaling in colonic CAFs, leading to increased tumor growth in vitro in 3D tumor spheroid assays

Next, we examined whether IL-1β treatment of CAFs affects tumor cell behavior by assessing growth and invasion of tumor cells in 3D spheroid assays (Fig. 4a, assay A). Tumor spheroids treated with conditioned media from CAFs pre-treated with IL-1β grew larger than those treated with conditioned media from untreated CAFs (Fig. 4b). The addition of an IL-1-inhibiting antibody to the CAF pre-treatment step resulted in an inhibition of this effect, demonstrating that the CAFs induced tumor spheroid growth by secreting IL-1β (Fig. 4b). In addition, we carried out organotypic invasion assays (Fig. 4a, assay B), in which we co-cultivated tumor spheres together with pre-treated fibroblasts in a collagen matrix and measured their invasion and outgrowth over time. We found that the co-culture with IL-1β-pre-treated CAFs led to an increase in the outgrowth area of invading tumor spheroids, while the addition of anti-IL-1β to the CAF pre-treatment step abrogated this effect (Fig. 4c). Furthermore, treatment of these co-cultures with Anakinra or anti-IL-1β reduced the spheroid-outgrowth-enhancing effect of CAFs, while their effect on tumor spheres in the respective monocultures was negligible (Fig. 4d, Supplementary Fig. 5a).

Subsequently, to substantiate the clinical relevance of these in vitro findings, we used the TCGA bulk expression dataset which we split into *IL1R1*[hi] and *IL1R1*[lo] expressing samples and performed GSEA. Strikingly, patients with high *IL1R1* expression showed an increase in pathways relevant to tumorigenesis, including epithelial to mesenchymal transition (EMT), angiogenesis and oncogene signaling, as well as pathways linked to inflammation (Fig. 4e). To confirm that the observed gene expression difference is specifically due to the effect of *IL1R1*[hi] CAFs on tumor cells, we analyzed our RNA-seq data described before (Supplementary Fig. 3c) but focusing this time on the response of tumor cells grown with either *IL1R1*[lo] or *IL1R1*[hi] CAFs. We found that *IL1R1*[hi] CAFs induced a more profound response in the tumor cells than *IL1R1*[lo] CAFs (Fig. 4f). Of note, we were able to confirm the enrichment

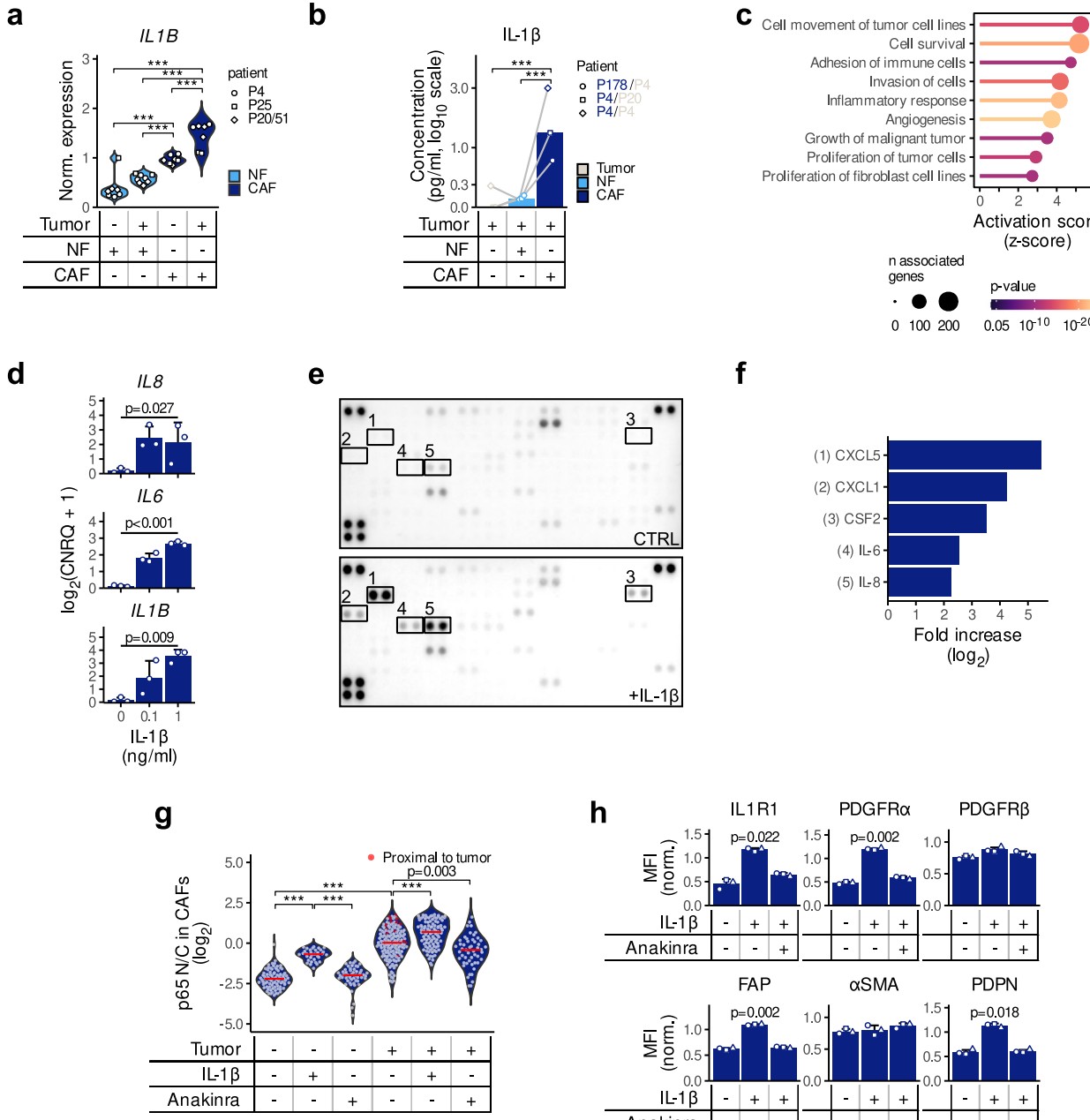

**Fig. 3 | Activation of the IL1-pathway in CAFs by tumor cells leads to an elevated secretion of pro-inflammatory cytokines. a** *IL1B* expression in fibroblasts upon co-culture with tumor cells quantified by qPCR. NF/CAF and tumor spheroid cultures were either paired (P4 and P25) or unpaired (P51 tumor and P20 NF/CAF). CNRQ expression values have been normalized by $log_2$ transformation followed by a non-centered scaling by patient pairs shapes (Tukey post-hoc test following a nested ANOVA design; ***$p < 0.001$). **b** IL-1β concentrations (pg/ml) in the media from tumor cells cultured alone or in the presence of NFs or CAFs. $n = 3$ independent experiments (connecting lines) from paired cultures (P4) or unpaired cultures (P178 tumor spheres and P4 NF/CAF) are shown (Tukey post-hoc test following a repeated measures ANOVA; ***$p < 0.001$). **c** Predicted activation of downstream cellular effects in CAFs upon IL1β stimulation. Differentially expressed genes were analyzed using the Ingenuity Pathway Analysis (IPA) software with the "Downstream Effects Analysis" function. **d** Expression of *IL8*, *IL6*, and *IL1B* after IL-1β stimulation (100 pg/ml or 1 ng/ml) of P4 CAFs. Log2 transformed CNRQ expression values from $n = 3$ independent experiments are shown as mean ± SD (Repeated

measures ANOVA). **e, f** Cytokine secretion triggered by IL-1β stimulation (1 ng/ml) in tumor fibroblasts (CT5.3 cells). The Proteome Profiler Human XL Cytokine Array Kit (R&D Systems) was used to identify cytokines secreted into the conditioned media ($n = 2$ independent experiments). Images in **e** were analyzed using ImageJ and the integrated densities in **f** were normalized to the values measured in the control condition (CTRL). **g** p65 nuclear-to-cytoplasmic ratio (N/C) in CAFs. After treating tumor cell (LS174T)−CAF (CAF-8) co-cultures with either IL-1β or Anakinra, ICC staining of p65 was quantified using ImageJ and N/C was calculated. Red dots show CAFs in close proximity to tumor spheroids (<25 μm). Tukey post-hoc test following a one-way ANOVA (***$p < 0.001$) ($n = 2$ independent experiments). **h** MFI of IL1R1, PDGFRα, PDGFRβ, FAP, αSMA and PDPN on IL-1β treated CT5.3 cells as assessed by flow cytometry. Values measured in $n = 3$ independent experiments were normalized (non-centered scaling) and bar charts represent the mean ± SD (Holm's adjusted two-sided pairwise paired *t*-test). Source data are provided as a Source Data file.

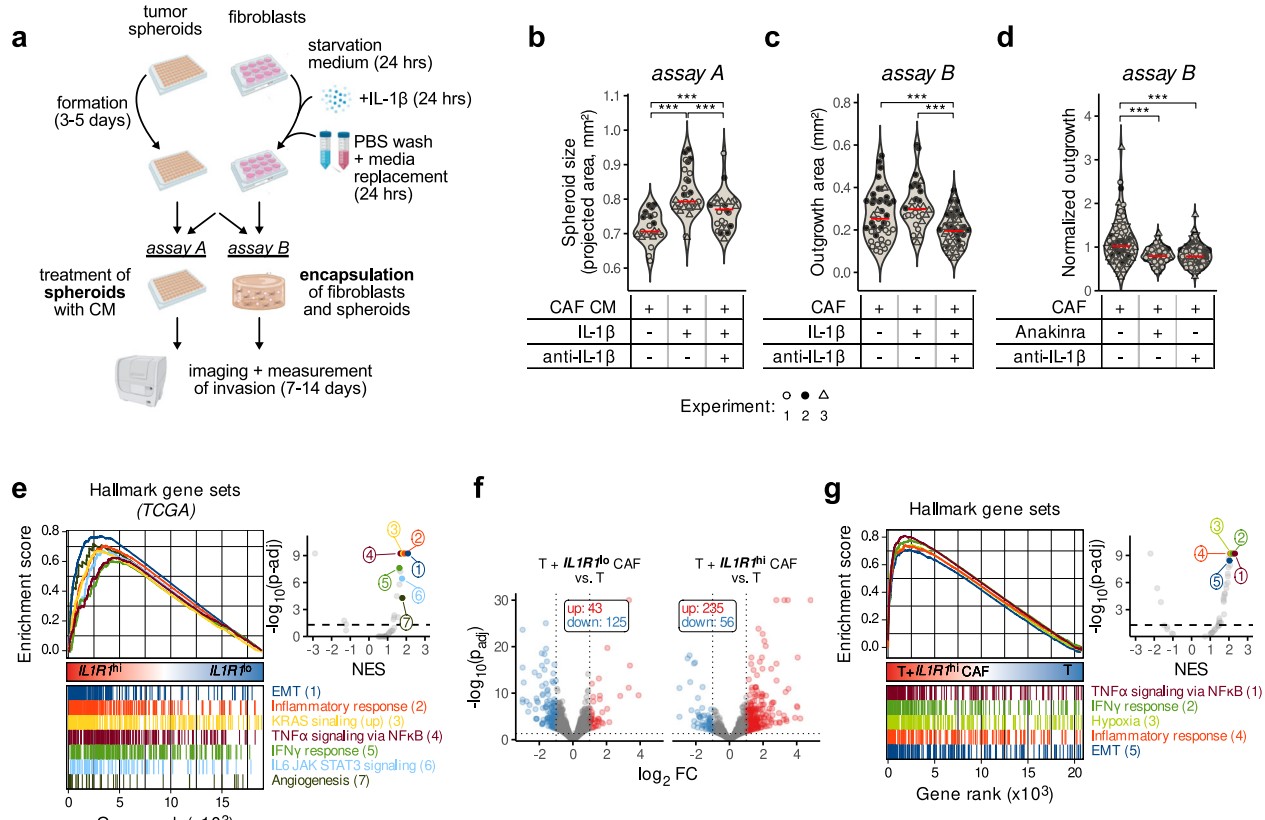

**Fig. 4 | IL-1β-activated CAFs promote tumor growth. a** Experimental layout of the spheroid growth (*assay A*) and collagen gel 3D organotypic assay (*assay B*). **b** Tumor epithelial cells (P4) were grown as spheroids (*assay A* in **a**) and treated with conditioned media (CM) collected from CAF cultures (P42) pretreated with either IL-1β (1 ng/ml) alone or in combination with anti-IL-1β (100 ng/ml). Once pretreated, cells were washed with PBS to remove any residual IL-1β and the medium conditioned during 24 h. Measures from 3 independent experiments after 9 days of growth are shown as different data point shapes (Tukey post-hoc test following a nested ANOVA design; ***p < 0.001). **c** Organotypic 3D assay of tumor spheroids (P4) cocultured with CAFs in collagen I gels (P42; *assay B* in **a**) pre-treated with either IL-1β (1 ng/ml) alone or in combination with anti-IL-1β (100 ng/ml). Measures from three independent experiments are shown as different data point shapes (Tukey post-hoc test following a nested ANOVA design; ***p < 0.001). **d** Organotypic encapsulation assay (*assay B* in **a**) where tumor spheroids (P4) were encapsulated with CAFs (CT5.3) and treated with anti-IL-1β (100 ng/ml) or Anakinra (100 ng/ml). The outgrowth areas from *n* = 4 independent experiments were

normalized (non-centered scaling by experiment). Statistically significant differences were determined using a nested ANOVA followed by Tukey's post-hoc test (***p < 0.001). **e** GSEA in *IL1R1*^hi (upper quartile, *n* = 128) vs *IL1R1*^lo (lower quartile, *n* = 128) TCGA tumors using the Hallmark gene sets. The running enrichment scores (ES) of most significant pathways of interest are shown in the left panel, and the volcano plot (right panel) shows the normalized enrichments scores (NES) against the log$_{10}$ of adjusted *p*-value. **f** Volcano plot showing the genes differentially expressed (RNA-Seq data) in HT-29 tumor spheroids when cocultured with either *IL1R1*^lo (*n* = 3 independent biological replicates from three different patients P16, P19, P22) or *IL1R1*^hi (*n* = 3 independent biological replicates from three different patients P32, P41, P42) CAFs. **g** MSigDB Hallmark GSEA in tumor spheroids (HT-29) upon co-culture with *IL1R1*^hi CAFs. The running enrichment scores (ES) for selected gene sets are shown in addition to the volcano plot showing the normalized enrichment scores (NES) and adjusted *p*-values of all 50 MSigDB Hallmark gene sets. Source data are provided as a Source Data file.

of genes involved in EMT and inflammation signaling by GSEA, when tumor cells were grown in presence of *IL1R1*^hi CAFs compared to tumor cell monocultures (Fig. 4g). Such an enrichment could not be detected when tumor cells were grown with *IL1R1*^lo CAFs (Supplementary Fig. 5b). Our observations suggest that tumor cells induce and maintain the autocrine IL-1β activation loop in CAFs, which altogether favors tumor progression. However, the exact condition and phenotype of tumor cells that can induce the autocrine loop in CAFs needs further investigation.

## IL1R1⁺ iCAFs are linked to an immunosuppressive TME in CRC

CAFs play an important role in shaping the immune environment of a tumor[30]. To determine how the immune system could be modulated by IL1R1⁺ iCAFs, we further subclustered tumor myeloid and T cells of the Lee and Qian scRNA-Seq datasets to identify macrophages (Supplementary Fig. 6a), CD4⁺ and CD8⁺ T cells as well as Tregs (Supplementary Fig. 6b). We then analyzed cell-cell communication in the TME using the ligand-receptor (LR) analysis framework LIANA[31]. Most of the

potential LR interactions detected were fibroblast autocrine signaling pathways (Supplementary Fig. 6c, d). However, when considering communications of IL1R1⁺ iCAFs with other cellular compartments, LIANA identified links with immune cells (macrophages, Tregs, CD4- and CD8-expressing T cells) in addition to epithelial cells (Fig. 5a and Supplementary Fig. 6e) and suggested that T cells might be regulated by IL1R1⁺ iCAFs (Fig. 5a, Supplementary Fig. 6e and Supplementary Table 4). Among the potential signaling pathways, we found CD69 signaling, which has previously been described in immune tolerance[32,33]. To further confirm whether CAFs with an active IL-1 pathway can modulate T cells, we stimulated mouse fibroblasts with either IL-1β alone or in combination with an anti-IL-1β antibody and subsequently co-cultured them with gp33-specific T cells (Fig. 5b). IL-1β-stimulated fibroblasts reduced the proliferative capacity of co-cultured gp33-activated T cells, and this effect was abrogated by the addition of the anti-IL-1β antibody (Fig. 5c).

In addition, LIANA highlighted potential LR communications between IL1R1⁺ iCAFs and macrophages and identified the IL-1β-IL1R1

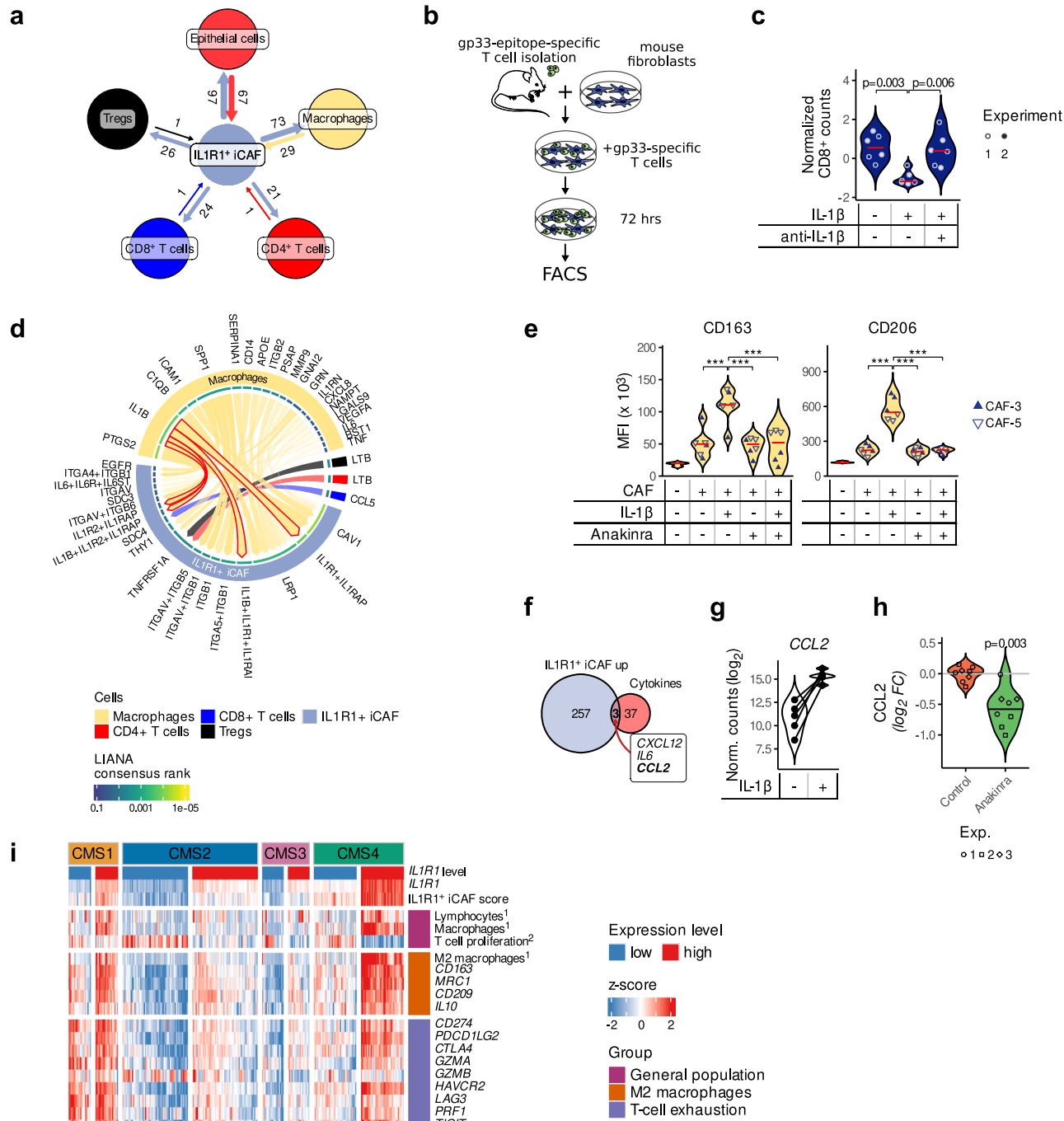

**Fig. 5 | IL1R1⁺ iCAFs regulate immune cells in the tumor microenvironment.**
**a** Potential LR interactions between IL1R1⁺ iCAFs and epithelial cells, macrophages and T cells detected by LIANA in the Lee scRNA-seq dataset. **b** Experimental setup of a CD8⁺ T cell proliferation assay. Gp33-specific T cells were isolated from spleens and lymph nodes of P14 TCRVα2Vβ8 mice and co-cultured with L cells (mouse fibroblast cell line). **c** Proliferation of CD8⁺ T cells after 72 h of co-culture with fibroblasts pretreated with IL-1β (1 ng/ml) alone or in combination with anti-IL-1β (100 ng/ml) as analyzed by flow cytometry. Data from two independent experiments (shown as filled and empty circles) are represented. Statistically significant differences were determined using a nested ANOVA followed by Tukey's post-hoc test. **d** Chord diagram showing LR interactions (LIANA aggregate score <0.05) between ligands borne by immune cells (macrophages, CD4⁺ T cells, CD8⁺ T cells and Tregs) and receptors borne by IL1R1⁺ iCAFs in the Lee scRNA-seq dataset. Arrow thickness and opacity shows higher ranked LIANA scores. Arrows outlined in red highlight the IL-1β-IL1R1 pair. **e** CD163 (left) and CD206 (right) expression on PBMC-

derived macrophages upon 48-h-long co-culture with IL-1β (1 ng/ml)−and/or Anakinra (100 ng/ml) pre-treated CAFs (n = 2 independent experiments with 2 different CAF cultures, CAF-3 or CAF-5, in each experiment). **f** Venn diagram showing the overlap of genes upregulated in IL1R1⁺ iCAFs and the geneset of secreted cytokines known to exert macrophage/monocytes chemotaxis. **g** Violin plot showing the expression of *CCL2* upon stimulation of fibroblasts with IL-1β (n = 5 independent samples each representing one patient-derived CAF culture). **h** Violin plot showing the secreted levels of CCL2 in CAFs upon Anakinra treatment. Data from three independent experiments are represented. **i** Expression of immune-cell-related markers and scores in TCGA CRC patients. Patients were divided into *IL1R1*^hi (upper quartile, n = 128) vs *IL1R1*^lo (lower quartile, n = 128) within each of the four CMS. The expression of *IL1R1* and immune-related genes, as well as IL1R1⁺ iCAF, immune infiltration (Thorsson et al., highlighted as ref. 1) and T cell proliferation (Szabo et al., highlighted as²) scores are shown (as z-scores). Source data are provided as a Source Data file.

interaction to be among the top candidates (Fig. 5d and Supplementary Fig. 6f). Indeed, in the scRNA sequencing datasets macrophages were the main *IL1B* expressing cells in the TME (Supplementary Fig. 1g), suggesting that IL-1β released by macrophages supports IL-1 signaling in CAFs in addition to the autocrine loop we described earlier. This privileged IL-1β-IL1R1 link between macrophages and IL1R1+ iCAFs led us to investigate whether the IL-1β signaling triggered in CAFs could also push tumor-associated macrophages (TAMs) to a tumor-promoting (*i.e.* an M2-like) phenotype. To this end, we co-cultured macrophages with CAFs and quantified the expression of M2-relevant markers by FACS. CD206 and CD163 expression remained unchanged when macrophages were directly treated with IL-1β (Supplementary Fig. 6g). However, when macrophages were cultured together with CAFs, which were pre-treated with IL-1β, CD163 and CD206 were significantly increased compared to controls co-cultured with untreated CAFs or when IL1R1 signaling was inhibited (Fig. 5e). To identify the potential CAF-derived cytokine/chemokine capable to polarize macrophages to a pro-tumorigenic (M2-like) phenotype, we overlaid the genes upregulated in IL1R1+ iCAFs with a gene set of secreted cytokines known to exert macrophage/monocytes chemotaxis. We identified three potential candidates: CCL2, CXCL12 and IL6 (Fig. 5f). While CXCL12 was not increased after IL-1β simulation (Supplementary Fig. 7a), CCL2 and IL6 were both up-regulated following treatment with IL-1β (Fig. 5g and Supplementary Fig. 7b). Additionally, IL6 and CCL2 were induced in CAFs upon co-culture with patient matched tumor organoids to a greater extent than in NFs along with the activation of NFκB target genes (Supplementary Fig. 7c). Importantly, we have previously shown that CRC CAFs, in coculture with tumor cells and monocytes/macrophages, substantially induce CCL2 in both CAFs and macrophages—a process dependent on CAF derived M-CSF expression—and thus generate an enhanced monocyte recruiting microenvironment[34]. Accordingly, in our experiments, Anakinra treatment in CAFs led to a significant decrease of CCL2 in CAFs (Fig. 5h). Altogether, this data suggests that IL1R1+ iCAFs may act in a pro-tumorigenic manner by suppressing immune cell responses. Confirming this, we also found that the expression of PD-L1/*CD274* and PD-L2/*PDCD1LG2* were both increased in cultures of CAFs when compared to paired NFs derived from three independent CRC patients (Supplementary Fig. 7d). IL-1β stimulation further increased PD-L1 expression at the mRNA (Supplementary Fig. 7e) and protein (Supplementary Fig. 7f) levels in patient-derived CAFs.

When examining immune cell infiltration scores in TCGA CRC patients across the four CMS subtypes, we observed that tumors expressing high *IL1R1* displayed an increased presence of immune cell populations, such as macrophages and lymphocytes. Strikingly, we observed that proliferation markers of T cells were reduced and that the expression of exhaustion markers, such as *LAG3*, as well as immunoregulatory proteins such as PD-L1/*CD274* and PD-L2/*PDCD1LG2*, was increased, in particular in CMS4 patients that expressed high levels of *IL1R1* (Fig. 5i). We similarly found increased expression levels of markers, which are characteristic of tumor-promoting M2 macrophages, in *IL1R1*hi tumors (Fig. 5i).

### IL1R1-expressing CAFs play a role in enhancing tumor growth in vivo

To evaluate the role of stromal IL-1-signaling in CRC in vivo, we carried out xenograft experiments in ColVIcre+IL1R1fl/fl mice (Fig. 6a,b). ColVIcre has been used to generate genetic stromal-compartment specific alterations[35] and using the mouse scRNA-Seq dataset GSE134255 we were able to confirm the preferential expression of *Col6a1* in fibroblasts, as previously described[35–37] (Supplementary Fig. 8a-c). Although scRNA sequencing datasets revealed endothelial cells as *IL1R1* expressing cells (Fig. 1b, Supplementary Fig. 1g, Supplementary Fig. 8a,c), the absence of *Col6a1* expression in these cells ensures the fibroblast specific knock-out of the receptor (Supplementary Fig. 8a-c).

Nevertheless, deciphering the relevance of IL1R1 signaling in endothelial cells needs to be considered in the future. First, we confirmed the loss of *IL1R1* expression in colon (Fig. 6c) and skin (Supplementary Fig. 7d) fibroblast cultures established from ColVIcre+IL1R1fl/fl mice. We then subcutaneously injected CRC cells into both flanks of these mice (Fig. 6b), a model in which we were able to observe infiltrating fibroblasts (Supplementary Fig. 8e). We observed better survival (Supplementary Fig. 8f) and lower tumor volumes (Fig. 6d,e, Supplementary Fig. 8g) in ColVIcre+IL1R1fl/fl compared to their littermate ColVIcre-IL1R1fl/fl control mice, indicating that an IL-1-related crosstalk between CAFs and tumor cells plays a cancer-promoting role. Additionally, when IL1R1 is absent in fibroblasts, fibroblasts tend to adopt a more myofibroblastic phenotype characterized by lowered expression of FAP and PDPN and upregulated expression of αSMA, PDGFRα and PDGFRβ (Supplementary Fig. 8H). We then analyzed different immune cell populations in the resected xenograft tumors. While we didn't detect a difference neither in the macrophage population (Supplementary Fig. 8i and Supplementary Fig. 9a) nor in the total number of T cells within the TME (Supplementary Fig. 8j, k and Supplementary Fig. 9b), an observation which might be due to the time point we chose, we observed a decrease in Th17 cells (Fig. 6f), which are majorly described as pro-tumorigenic[38], in ColVIcre+ IL1R1fl/fl mice compared to their littermate controls. IL-1β has previously been reported to be involved in Th17 differentiation[39,40]. Along this line, when co-culturing naive T cells with IL1R1-/- fibroblasts (derived from ColVIcre+ IL1R1fl/fl mice), we observed a reduced differentiation into Th17 cells when compared to co-culture with fibroblasts isolated from control mice (Fig. 6g and Supplementary Fig. 10a). Finally, we analyzed the expression of PD-L1 as an overall marker for an immunosuppressive TME. The quantification of PD-L1 staining derived from these mice showed a lower expression in tumors from ColVIcre+ IL1R1fl/fl mice than those from the respective control mice (Fig. 6h), suggesting that IL1R1-positive CAFs induce immunosuppression in the TME, similarly to what we observed in *IL1R1*hi patients (Fig. 5i). To further underline the potential clinical application of our findings, we treated tumor-bearing mice with Anakinra (Fig. 6b, lower part). Strikingly, and in agreement with the results obtained in the ColVIcre+IL1R1fl/fl mice, we observed a reduction in tumor growth upon IL1R1 blockade (Fig. 6i,j). Importantly, similarly to ColVIcre+IL1R1fl/fl mice, we observed less IL-17+ cells in tumors from Anakinra-treated mice (Fig. 6k and Supplementary Fig. 10b), further supporting the notion that CAFs may play their tumorigenic role by secreting IL-1β-driven factors able to attract differentiated Th17 cells to the TME. Finally, we confirmed these findings on CRC patients using the TCGA dataset. After dividing the cohort into IL1R1hi- and IL1R1lo-expressing tumors (lower and upper quartiles in either all CMS or in CMS4), we observed that a high expression of *IL1R1* was associated with an increase in the expression of genes linked to Th17 cells (Fig. 6l, Supplementary Fig. 8l).

## Discussion

Here, we highlight the presence of a CAF subtype, IL1R1+ iCAF, which is characterized by increased IL-1β-signaling and elevated expression of *IL1R1* and acts in a pro-tumorigenic manner both in vitro and in vivo. We were able to correlate the expression of *IL1R1* with IL1R1+ iCAFs markers, such as *FAP* and *CXCL12*. Furthermore, the expression of *IL1R1* is enriched in CMS4 CRC patient samples and predictive of lower survival. Specifically ablating the IL-1 pathway in fibroblasts reduces tumor growth in 3D organotypic assays and tumor volumes in vivo. The latter may be potentially influenced by the reduction of the infiltration of pro-tumorigenic Th17 cells and by reversing the CAF-induced immunosuppression in the TME. Altogether, our results indicate the importance of stromal IL-1 signaling in tumor development (Fig. 7).

Interestingly, some of the genes associated with IL1R1+ iCAFs possess functionalities that may reinforce their pro-tumorigenic

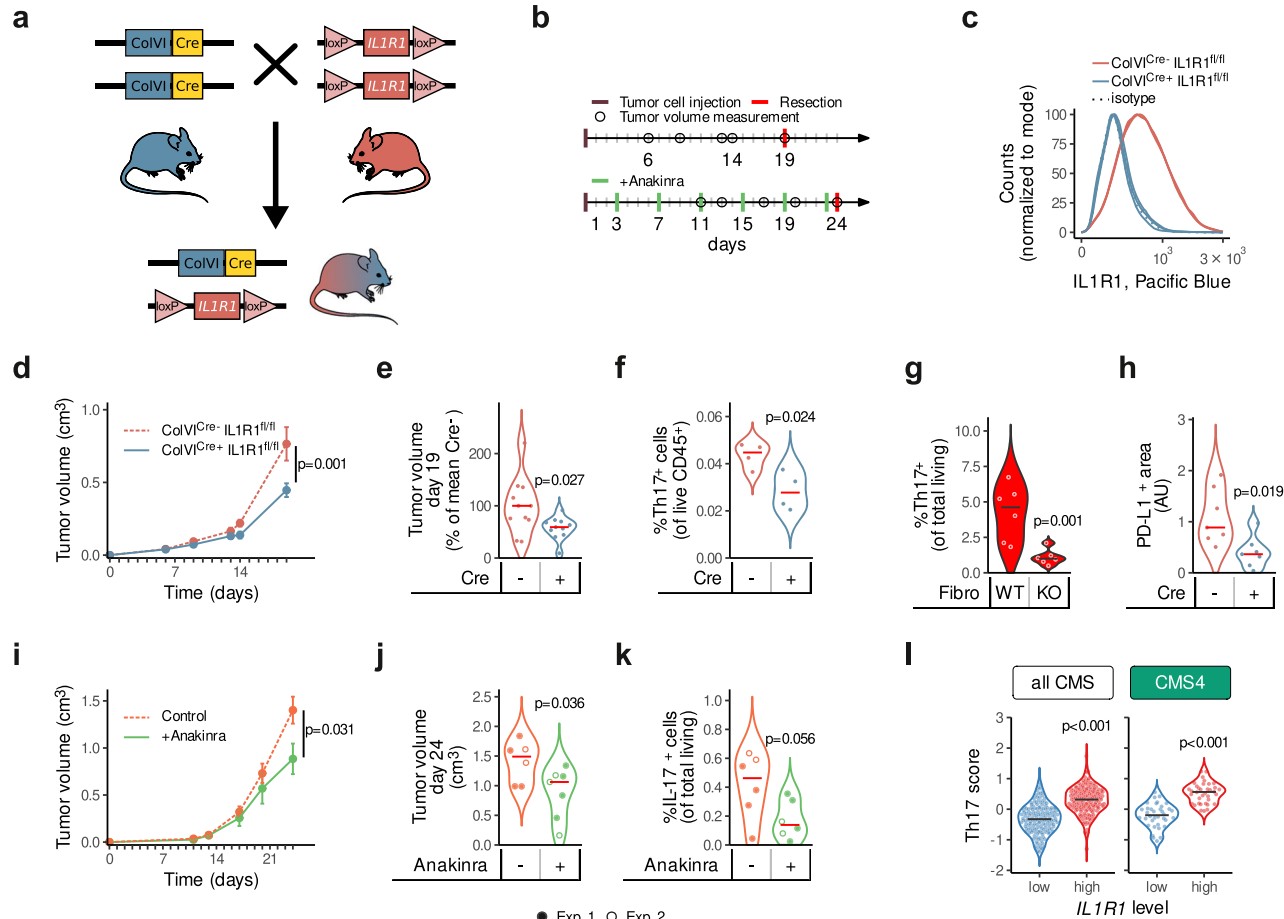

**Fig. 6 | IL1R1 ablation or inhibition in vivo reduces tumor growth and regulates immune cells in the TME. a** Generation of ColVI^cre+^IL1R1^fl/fl^ mice. **b** Experimental design of both in vivo assays. *Upper diagram*: MC38 cells were injected into both flanks of ColVI^cre+^IL1R1^fl/fl^ mice and ColVI^cre-^IL1R1^fl/fl^ control littermates and tumor growth was measured over time. *Lower diagram*: CT26 cells were injected into both flanks of BALB/c mice. Mice were treated with Anakinra (10 mg/kg) on days 3, 7, 11, 15 and 23 after tumor cell implantation and tumor growth followed over time. **c** IL1R1 expression in colon fibroblasts from ColVI^cre+^IL1R1^fl/fl^ and ColVI^cre-^IL1R1^fl/fl^ mice as measured by FACS, n = 3 mice per condition. **d–f**. Tumor volumes (cm³) measured over time and shown as mean ± SEM from one representative experiment (*n* = 4 mice in the ColVI^cre+^IL1R1^fl/fl^ and *n* = 5 mice in the ColVI^cre-^IL1R1^fl/fl^ condition, respectively) out of three independent experiments in **d**, tumor volumes at experimental endpoint pooled from three independent experiments and normalized to the control, with *n* = 11 mice per condition in **e**, Th17 cells (CD4⁺ RORγT⁺ IL-17⁺ cells, shown as % of live cells) in tumors as assessed by FACS in one representative experiment of three, with *n* = 4 mice per condition in **f** in ColVI^cre+^IL1R1^fl/fl^ and ColVI^cre-^IL1R1^fl/fl^ mice subcutaneously implanted with MC38 cells. **g** Th17 cell differentiation (CD4⁺ RORγT⁺ IL17⁺ cells, shown as % of live cells) upon co-culture with fibroblasts from

ColVI^cre+^IL1R1^fl/fl^ (KO) or ColVI^cre-^IL1R1^fl/fl^ (WT) mice (*n* = 6 per condition), as assessed by FACS. **h** PD-L1 expression in tumors assessed by IF in one representative experiment of three, with *n* = 7 tumors per condition in ColVI^cre+^IL1R1^fl/fl^ and ColVI^cre-^IL1R1^fl/fl^ mice subcutaneously implanted with MC38 cells. **i–k** Tumor volumes (cm³) measured over time and shown as mean ± SEM with *n* = 8 mice treated with Anakinra or *n* = 7 mice treated with the vehicle control out of two independent experiments in **i**, tumor volumes at experimental endpoint pooled from two independent experiments in **j**, IL-17-producing cells (shown as % of live cells) isolated from tumors at experimental endpoint as assessed by flow cytometry in **k** in mice treated with Anakinra or the vehicle control. Two independent experiments are pooled in j-k to show a total of *n* = 7 mice treated with Anakinra or *n* = 6 mice treated with the vehicle control. Volumes of tumors from both flanks were averaged for each mouse in panels **d**, **e**, **i** and **j**. Th17 and IL-17⁺ cells from both flanks were averaged for each mouse in **f** and **k**. **l** Th17 scores in TCGA patients. Patients were split into *IL1R1*^hi^ (upper quartile) and *IL1R1*^lo^ (lower quartile), either in all CMS or in CMS4 only. Repeated measures ANOVA in **d** and **h**. Two-sided unpaired *t*-tests in **e–g**, **i–l**; ***p < 0.001. Red or black horizontal lines in **e–h** and **j–k** represent the median.

function. For example, *RSPO3* (Fig. 2d) is a well-known activator of the WNT/β-catenin pathway. It has been linked to increased tumor growth and tumorigenicity in Wnt-mutated colorectal tumors[41] and is a mediator of invasiveness in prostate cancer[42], indicating that IL1R1⁺ fibroblasts may also play a role in Wnt signaling. In addition, genes such as *IGF1*[43] have been associated with increased CRC tumor growth and aggressiveness, suggesting that the IL1R1⁺ iCAF subtype may possess other potential pro-tumorigenic effects in addition to its elevated *IL1R1* expression and IL-1β signaling.

Recent research has also highlighted that certain CAF subtypes play immunomodulatory roles, such as in PDAC[11,44]. We show that the IL1R1^hi^ subtype is associated with immune cell recruitment and function, in particular the Th17 cell population. Although Th17 cells have

been described as having both tumor-promoting and tumor-reducing functions, most studies highlight a tumor-supportive effect[38]. For example, it was reported that tumor growth is reduced in *IL17*^-/-^ mice, probably due to the lost capacity of IL-17 to promote tumor growth via the IL-6/STAT3 pathway[45]. Additionally, high Th17 cell infiltration has been linked to poor prognosis in patients[46]. According to a previous study, fibroblast-derived factors, including IL-6 and IL-8, might attract Th17 cells to the tumor niche[47]. Thus, cytokines secreted by IL-1β-activated CAFs, including IL-1 itself, IL-6 and IL-8, might favor the differentiation and proliferation of Th17 cells, as well as their recruitment to the tumor niche, which would ultimately favor tumor progression. We also observed higher levels of common immune cell exhaustion markers and genes linked to immunotherapy efficacy, such as PD-L1/

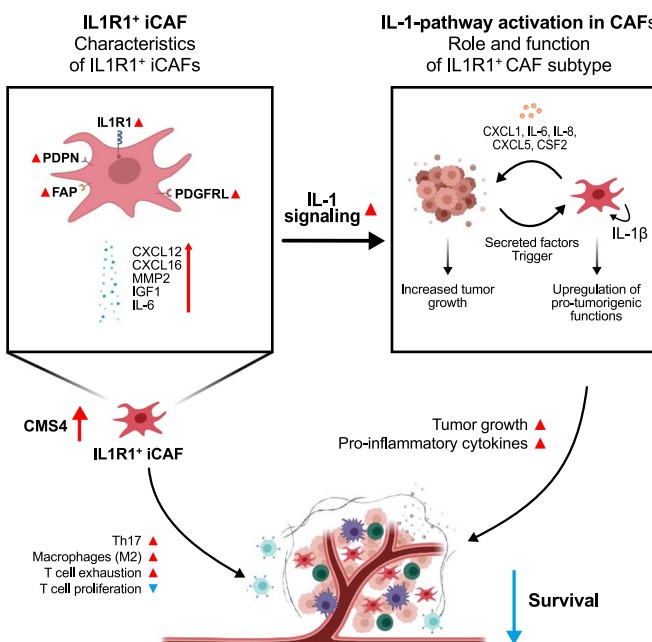

**Fig. 7 | A summary of the roles played by the IL1R1⁺ CAFs in the CRC TME. The IL1R1⁺ subtype of CAFs is linked to lower survival and to higher tumor growth and immune evasion (M2 polarization and T cell exhaustion markers).** IL1R1⁺ iCAFs show increased IL-1 signaling and NFκB pathway activation, leading to the secretion of pro-inflammatory cytokines participating in tumor growth. Finally, IL1R1⁺ CAFs favor infiltration of Th17 cells into the TME.

CD274 and PD-L2/PDCD1LG2. Interestingly, recently published research shows that tumoral IL-1β production and regulation, as well as inflammatory responses, are upregulated in αSMA⁺-CAF-depleted mice and that these mice display increased immunosuppression[48]. This suggests that myCAFs negatively regulate iCAFs and that the depletion of ACTA2⁺ fibroblasts leads to increased proliferation of IL-1β-driven iCAFs. Consequently, the lack of myCAFs would lead to IL1R1⁺ iCAF driven immunosuppression and lowered CD8⁺ T-cell proliferation as observed in our study. Of note, simultaneously targeting *FAP* and *CXCL12*, two genes which we found to correlate with *IL1R1* expression in patient datasets, has been shown to act synergistically with anti-PD-L1 therapy in vivo[49,50]. Future studies will be needed to explore the potential of combinatorial effects of IL-1 blockade with checkpoint inhibitors.

Another interesting aspect worth considering is the potential role that IL-1 signaling may play in the conversion of normal fibroblasts to pro-tumorigenic fibroblasts. Recent publications have highlighted the capacity of CAFs to "re-educate" normal fibroblasts into tumor-enhancing fibroblasts in an NFκB dependent manner[51]. In our data, we show that tumor cells induce IL-1 signaling in surrounding fibroblasts, suggesting that a similar "education" pathway may be involved. While there are other major sources of IL-1, such as macrophages[52], that may play a significant role in later stages of tumor development, reprogramming of local fibroblasts into more pro-tumorigenic IL-1 phenotypes may serve to drive early tumor development.

Collectively, our study highlights the importance of IL1R1 and IL-1β signaling in the fibroblast compartment, especially in regard to driving immunosuppression and tumor growth. Therefore, targeting this population may be a potential therapeutic option to support traditional anticancer therapies. Whether this is achieved via specifically targeting IL-1 signaling via anti-IL-1 antibodies or antagonists, such as Anakinra, or targeted deletion of IL1R1⁺ subtypes of fibroblasts in the TME (for example via linked markers such as PDPN) requires further research, but the potential for more efficient therapies is evident,

especially for patients with hard-to-treat, highly stromal subtypes of colorectal cancer.

## Methods

### Patient sample collection and preparation
All patient samples were obtained through our established colorectal cancer cohort and handled according to all institutional guidelines and regulations (ERP-16-032), based on previously described biospecimen handling standards[53].

All patient samples were donated freely, with informed consent and with the full cooperation of each patient. Ethical approvals were obtained for samples collected in Luxembourg and Vienna from the Comité National d'Ethique de Recherche (CNER) (Reference 201009/09) and the ethics committee of the Medical University of Vienna (EK Nr: 1580/2017), respectively.

### Cell cultures
Primary tumor samples obtained from patients were washed several times in Dulbecco's Modified Eagle Medium (DMEM/F12) supplemented with an antibiotic-antimycotic agent (15240062, Gibco), then cut into 1–2 mm³ pieces and incubated with Collagenase IV (0.05 mg/ml; Invitrogen) and hyaluronidase (2 μg/ml; Sigma) for 1 h at 37 °C. The resulting suspension was filtered using a 70 μm cell strainer (Greiner Bio-One) and either transferred into ultra-low attachment (ULA, Corning) flasks for epithelial spheroid cultures or to petri dishes (∅ 10 cm) for fibroblasts. Fibroblasts were allowed to adhere for 20 min before the media was transferred to a secondary petri dish and replenished. Fibroblasts were cultured in DMEM-F12, supplemented with 10% fetal bovine serum (FBS) and 1% penicillin/streptomycin (P/S, 1×, Lonza), for 24 h, before being replaced by Endothelial Growth Medium 2 (C-22010, Promocell) for long term culture. Epithelial spheroid cultures were cultivated in serum-free stem cell medium (SCM) composed of DMEM-F12 supplemented with B-27 (1×), N-2 (1×, Invitrogen), bovine serum albumin (4 mg/ml, Roth), non-essential amino acids (1×, Sigma), glucose 0.15% (Sigma), Insulin (4 U/l, Sigma), Heparin (4 μg/ml), N-Acetylcysteine (1 mM, Sigma), endothelial growth factor (20 ng/ml, Biomol), basic fibroblast growth factor (20 ng/ml, Miltenyi Biotec), and P/S (1×, Lonza).

CAF05 were purchased from Neuromics and maintained in endothelial growth media. Immortalized primary human colon fibroblasts (CT5.3) were graciously provided by Prof. Oplivier De Wever and were cultivated according to previously established protocols[54]. Non-transduced primary fibroblasts were grown in Endothelial Growth Medium 2-MV (C-22010, Promocell) before being transferred to DMEM-F12 (10% FBS, 1% P/S) for experimental assays. MC38 (mouse tumor cells) were obtained from Kerafast (ENH204-FP) and cultured according to the supplier protocol. CT26 (CRL-2638, mouse tumor cells) and L-cells (CRL-2648, normal mouse fibroblasts) were obtained from ATCC and were cultured as indicated by the provider. All cells were recently authenticated by STR profiling.

### In vivo assays
Animal experiments were performed according to all applicable laws and regulations, after approval by the Animal Experimentation Ethics Committee (AEEC) of the University of Luxembourg, as well as by the Luxembourgish Ministry of Agriculture, Viticulture & Rural Development (LUPA 2020/10) in compliance with the EU Directive 2010/63/EU, as well as the Grand-Ducal Regulation of 11 January 2013 (2013-1-11) on the protection of animals used for scientific purposes. These included the justification of the use of animals, their welfare and the incorporation of the principles of the "3Rs" (Replacement, Reduction and Refinement). The animals were maintained under standard habitat conditions (humidity: 40–70%, temperature: 22 °C) with a 12:12 light cycle. All experiments were carried out with veterinarian consent and all animal protocols were reviewed by a biostatistician. Mice of both

genders with the age between 8–12 weeks were used. In the in vivo experiments presented in Fig. 6, humane endpoint (HEP) is defined by the AEEC as the combined size of tumors on both flanks not surpassing 2000 mm³. This limit was not exceeded in our experiments.

**_Il1r1_ conditional knock-out mouse model.** ColVI<sup>cre+</sup> mice were a kind gift from Dr. George Kollias[36] and IL1R1<sup>fl/fl</sup> mice were obtained from The Jackson Laboratory. Both mouse lines were maintained on the C57Bl/6 J background and further crossed in house to establish ColVI<sup>cre+</sup>IL1R1<sup>fl/fl</sup> mice. MC38 tumor cells were prepared for subcutaneous injection in GFR (Growth Factor Reduced) Matrigel (354230, Corning) and injected at 500,000 cells per flank into ColVI<sup>cre+</sup>IL1R1<sup>fl/fl</sup> and littermate ColVI<sup>cre-</sup>IL1R1<sup>fl/fl</sup> control mice. Tumor volumes were measured until the humane endpoint or when the experimental endpoint was reached, upon which mice were euthanized via cervical dislocation and tumors were resected for further analyses.

**Pharmacological inhibition mouse model.** BALB/c mice were obtained from Charles River Laboratories. CT26 tumor cells were prepared for subcutaneous injection in GFR Matrigel (354230, Corning) and 500,000 cells were injected into each flank of BALB/c mice. Anakinra (10 mg/kg) was administered intraperitoneally every 4 days. Tumor volumes were measured over time until reaching the humane endpoint or the experimental endpoint, upon which mice were euthanized via cervical dislocation and tumors were resected for further analyses.

**Ex-vivo immune phenotyping**
Collected tumors were digested using the Mouse Tumor Dissociation Kit (130-096-730, Miltenyi Biotec) according to the manufacturer's protocol and stained in 50 μl of FACS buffer (phosphate buffered saline (PBS, Gibco) with 1% FBS and 1 μM EDTA (Sigma)) for 30 min at 4 °C using the antibodies at the concentrations indicated in Supplementary Table 5. For the detection of intracellular cytokines, cells were restimulated with 0.1 μg/ml phorbol 12-myristate 13-acetate (PMA, P1839, Sigma) and 1.5 μg/ml ionomycin (J60628.X0, Alfa Aesar) for 4 h at 37 °C in the presence of Brefeldin A (1×, eBioscience™). Intracellular cytokine staining was performed using the Cytofix/Cytoperm™ Fixation/Permeabilization Kit (554714, BD Biosciences) according to the manufacturer's protocol and using the antibodies listed in Supplementary Table 5. Gating strategies to identify CD4⁺, CD8⁺, IL−17⁺ T cells as well as macrophages are provided in the Supplementary Figs. 9, 10.

**Real-time qPCR**
Total RNA from samples was extracted using the miRNeasy Mini Kit (Qiagen) followed by reverse transcription using the miScript II RT kit (Qiagen) or the High-Capacity cDNA Reverse Transcription Kit (Applied Biosciences). All procedures were carried out according to the manufacturer protocols. qPCR reactions were run on the CFX384 Touch Real-Time PCR Detection System using AbsoluteBlue qPCR SYBR Green Low ROX (Thermo Scientific). Primer sequences can be found in Supplementary Table 5. Expression levels (Ct values) of genes of interest were normalized against multiple housekeeping genes (_YWHAZ_ and _EEF1A1_) using qbase+ software (Biogazelle).

**ELISA**
IL−1β concentrations in cell culture supernatants were measured using the Human IL-1 beta/IL-1F2 DuoSet ELISA (DY201-05, R&D Systems) according to the manufacturer instructions. CCL2 concentrations in cell culture supernatants were measured using the Human CCL2/MCP-1 DuoSet ELISA (DY−279−05, R&D Systems) according to the manufacturer protocol. Absorbance was measured using the Cytation 5 cell imaging multi-mode reader (Biotek) at 450 nm, with a wavelength correction reading at 570 nm.

**Western blot**
Cells were trypsinized, washed with cold PBS and lysed with RIPA buffer on ice. Obtained samples were heated at 95 °C in Lämmli buffer and resolved on SDS-PAGE 12% gels. Sampleswere transfered onto a polyvinylidene difluoride–PLUS Transfer Membrane (0.2 μm). Membranes were blocked with 10% BSA for 1 h at room temperature, followed by an overnight staining with primary antibody (Phospho-NF-κB p65 (#3033 P) Cell Signalling Rabbit at 1:1000, NFκB p65 (sc-109) Santa Cruz Rabbit at 1:200, Actin (MAB1501) Millipore Mouse at 1:5000, IL-1β (#12242) Cell Signalling Mouse at 1:1000) at 4 °C. Horseradish peroxidase (HRP)-labelled secondary antibodies (7076 s and 7074 at 1:5000) were used for visualisation. The blots were revealed using enhanced chemoluminescence on a Fusion FX imaging platform.

**Co-culture assays**
**Co-culture of tumor spheres with CAFs.** Tumor spheres were dissociated and 250,000 to 500,000 cells were seeded per well into Ultra-Low Attachment six-well plates (Greiner Bio-One). Transwell inserts (3 μm pore size, ThinCert®, Greiner Bio-One) were inserted into the wells and CAFs were seeded on top of them at 250,000–500,000 cells per well. Plates were incubated at 37 °C for 72 h. After 72 h, the inserts were transferred into new six-well plates, washed with PBS (1×, Gibco) and fibroblasts were collected. For the tumor cell compartment, the media-suspended spheres in the bottom compartment were centrifuged at 300 g for 5 min and the supernatant was collected for further use. The fibroblasts and the pelleted tumor cells were lysed in 700 μl of Qiazol lysis reagent (Qiagen). Lysed cells were vortexed and then stored at −80 °C until use.

For spheroid growth assays, tumor spheres were formed by seeding 2500–5000 tumor cells per well in DMEM-F12 (10% FBS, 1% P/S) in a 96-well plate. Plates were centrifuged 200–250 g for 1 min, then incubated at 37 °C. Cells were grown for 3–5 days until formation of spheroids, then 100 μl of conditioned media (1:1 v/v) was added. Sphere surface area was measured over one to 2 weeks using the Cytation 5 cell imaging multi-mode reader (Biotek).

For the 3D collagen I gel assays, tumor spheres were formed three to 5 days before seeding, as described above. 200,000 to 500,000 fibroblasts were seeded into a six-well plate in starvation media (DMEM-F12, 1% FBS, 1% P/S) for 24 h, then cells were stimulated for 24 h as indicated. After stimulation, fibroblasts were washed with PBS (1×, Gibco), detached using trypsin, washed in FBS-containing media and counted. The spheres and fibroblasts were centrifuged at 300 g. Fibroblasts were resuspended in 330 μl of 2.5% collagen and then transferred onto the spheroids. The resulting mixture was then transferred into a silicon form, holding a pre-made polyethylene terephthalate ring (pore size: 100 μm) on a petri dish, and incubated at 37 °C for 3 min. The petri dish was flipped upside down for 1 min to avoid spheroid sinkage to the bottom of the hydrogel. After polymerization the silicone forms were removed, and the gels transferred into 24-well plates containing DMEM-F12 + 10% FBS in presence of IL-1β (1 ng/ml, 200-01B, Peprotech EC. Ltd) or/and anti-hIL-1β-IgG (hereafter noted anti-IL-1β, mabg-hil1b-3, Invivogen). Sphere outgrowth was measured for one to 2 weeks using the Cytation 5 cell imaging multi-mode reader (Biotek).

**Co-culture of macrophages with CAFs.** Monocytes were isolated from peripheral blood and treated with 20 ng/ml M-CSF every second day for a total of 6 days. 300,000 CAFs (CAF-3 or CAF-5) were plated in Petri dishes (∅ 6 cm) containing 2 ml of EGM-MV2. After 7 h, CAFs were starved during 24 h with DMEM-F12 containing 1% FBS and 1% of L-glutamine. CAFs were then treated during 24 h with either IL-1β (100 ng/ml), Anakinra (1 ng/ml) or a mix of both (Anakinra was added 1 h before IL-1β). On day 7, CAFs were washed 3 times with PBS and 300,000 macrophages seeded on top of CAFs in 2 ml of RPMI

containing 10% FBS and 1% L-glutamine. After additional 2 days, the expressions of CD163 and CD206 were assessed by flow cytometry.

## Bulk RNA sequencing
RNA libraries were sequenced as 100 bp paired-end runs on a HiSeq2500 System (Illumina). Differential expression was analyzed using DESeq2. Genes with an adjusted $p$-value <0.05, a fold change >1, and a mean expression value >50 counts were considered significant.

## scRNA sequencing
For single-cell sequencing, primary patient tumor samples were dissociated using the gentleMACS™ Octo Dissociator and the Human Tumor Dissociation Kit (130-095-929, Milteny Biotec) according to the protocol outlined by the manufacturer. The cells were then treated with Debris Removal Solution (130-109-398, Milteny Biotec), followed by hemolysis buffer for 2 min (12146.00250, Morphisto) before quenching with Roswell Park Memorial Institute Medium 1640 (RPMI, Gibco) + 10% FBS + 1% P/S (1×, Lonza). Dead cells were removed by magnetic cell sorting (Dead Cell Removal Kit, 130-090-101, Milteny Biotec) with the LS columns (130-042-401, Milteny Biotec) on the QuadroMACS Separator (130-091-051, Milteny Biotec). Live cells were counted using haemocytometers. Single cell RNA sequencing was performed as previously described[55]. Drop-Seq libraries were prepared and sequenced on a NextSeq 500 System (Illumina).

## Cytokine array
Primary fibroblasts were starved for 24 h, stimulated for 24 h, and then washed with PBS (1×, Gibco). Conditioned media was collected and analyzed using the Proteome Profiler Human XL Cytokine Array Kit (R&D Systems) according to the protocol. The resulting dot arrays were imaged using the Fusion FX imaging system (Vilber Lourmat) and optical densities were quantified using the ImageJ software.

## Staining and flow cytometry
Cells were dissociated into a single cell suspension, antibodies added (Supplementary Table 5) and acquisition was done on BD FACSCanto II or BD FACS LSRFortessa flow cytometers as previously described[34]. Gating strategies are provided in the supplementary Figs. 9, 10.

## FACS sorting
CT5.3 fibroblasts were dissociated into a single cell suspension and stained for 30 min at 4 °C with an anti-IL1R1 antibody (FAB269P, R&D Systems) and DAPI. 20% of viable cells with high and low expression of IL1R1 were sorted using a BD FACSMelody cell sorter and FlowJo 10.6.1 was used for analysis.

## T cell proliferation assay
The influence of the IL−1-activated CAFs on T cell proliferation was assessed using a murine T cell proliferation assay. Briefly, 100,000 mouse fibroblasts (L-control cells) were treated with 1 ng/ml IL-1β in the presence of 100 ng/ml of anti-IL1R1 or 100 ng/ml of control IgG. After 24 h, cells were washed with PBS (1×, Gibco) and loaded with the LCMV gp33−41 peptide (1 μg/ml, RP20257, BioConnect) for 2 h. Gp33−41-specific CD8+ T cells were isolated from splenocytes of P14 TCR transgenic mice (provided by Prof. D. Brenner) using immune-magnetic separation and labeled with carboxyfluorescein succinimidyl ester (CFSE). Labeled CD8+ T cells were co-cultured with gp33-loaded fibroblast cells in presence of murine IL-2. After 72 h, T cells were stained with an anti-CD8 antibody (558106, BD Biosciences) and the proliferation was measured by flow cytometry as described above.

## T cell in vitro differentiation assay
Naïve T cells were isolated from spleens of C57BL/6 J mice using Mouse Naïve CD4+ T Cell Isolation Kit (130-104-453, Milteny Biotec). $2 \times 10^5$ cells were added to 96-well plates, which were previously seeded with

$2 \times 10^4$ ColVI^cre+IL1R1^fl/fl or ColVI^cre-IL1R1^fl/fl colonic fibroblasts. Cells were cultured with 30 ng/ml IL−6 (130-094-065, Milteny Biotec), 2 ng/ml TGF-β (240-B-002, R&D Systems), 5 μg/ml anti-IFNγ (554408, BD Parmingen), 5 μg/ml hamster anti-CD3 (100340, BioLegend), 1 μg/ml hamster anti-CD28 (102116, BioLegend) antibodies and 2 μg/ml anti-hamster secondary antibody (6060-01, SouthernBiotech). After 72 h, cells were restimulated with 0.1 μg/ml PMA (P8139, Sigma) and 1.5 μg/ml ionomycin (J60628.X0, Alfa Aesar) for 4 h at 37 °C in the presence of Brefeldin A (1×, eBioscience™). Intracellular cytokine staining using the Cytofix/Cytoperm™ Fixation/Permeabilization Kit (554714, BD Biosciences) according to the manufacturer's protocol and using the antibodies listed in Supplementary Table 5.

## Data processing
**Cole dataset and CLZ integration.** The Cole scRNA-sequencing dataset count matrices were processed and analyzed in R using the Seurat package. Ribosomal genes were removed and only cells with features between 200 and 2000 and <15% of mitochondrial genes were kept. Data were lognormalized and scaled, with "nCount_RNA" and "percent.mt" regressed out. PCA reduction was performed using the 2000 most variable features, with harmony used to adjust for variation between patients. UMAP reduction was computed on the 40 first dimensions of the corrected components (resolution 0.8), and clusters were assigned to cell types using canonical markers[25] and the SingleR package using the BlueprintEncodeData as a reference. The CLZ dataset was integrated using Seurat, with datasets reduced to common genes and merged, normalized and scaled while regressing out "nCount_RNA" and batch effect removed using Batchelor[56]. Following parameters were used: min.cells = 3, min.features = 200 and cell types were assigned using "clustify()" from clustifyr package, based on the Cole cell types as a reference. Harmonised data were reduced using UMAP on the 40 first dimensions of harmonized PCA projections, with tumor fibroblasts subclustered based on gene expression. Clusters with high expression of *RGS5*, *CSPG4* and *ACTA2* were labelled as myCAFs and clusters exhibiting high cytokine expression levels—as cyCAFs. The latter were further subdivided into cyCAF-1 and cyCAF-2 based on the expression of *IL1R1* and *CXCL12* (Supplementary Fig. 2b).

**Lee dataset.** Raw UMI counts for the Lee dataset, composed of the SMC and KUL3 cohorts, were downloaded from the GEO portal (GSE132465 and GSE132465) and imported into Seurat with min.cells = 3 and min.features = 200. Cells were filtered based on nFeature_RNA > 200, Feature_RNA < 10000 and percent.mt < 15 were kept, then lognormalised and scaled with covariates "nCount_RNA", and "percent.mt" regressed out. PCA reduction was performed using the 2000 most variable features, and variation between both cohorts was corrected using harmony. Reduced data was further segmented in clusters, using the 30 first UMAP dimensions (resolution of 0.5). Clusters were assigned to cell types using list of canonical markers[25] and the SingleR package using the BlueprintEncodeData as a reference. Identified fibroblasts, myeloid cells and T cells from tumor tissue were further extracted and similarly subclustered as described above using the resolutions of 0.2 for fibroblasts and 0.5 for myeloid and T cells.

**Qian dataset.** Counts for the Qian dataset (CRC samples) were downloaded from the authors website (https://lambrechtslab.sites.vib.be/) and processed using the parameters provided by the authors whenever available. Data with min.cells = 10 and min.features = 200 were imported in Seurat. Cells with nFeature_RNA < = 6000 and percent.mt ≤ 15 were kept and further normalised and scaled, regressing out "nCount_RNA", "percent.mt", "S.Score" and "G2M.Score". Variable genes were determined on genes with ExpMean between 0.0125 and 3, and dispersion >3. Data were reduced using PCA and UMAP (using the 40 first PC). Segmentation of the reduced data, cell type identification

and subclustering of tumor firbroblasts, myeloid and T cells were performed as described for the Lee dataset.

**Integration of CLZ, Lee and Qian.** The five cohorts Cole, Zhang, KUL3, SMC and QIAN (the Li cohort was not retained given the lower number of cells and overall quality) were normalized independently using the "SCTransform()" function provided by Seurat with the "vst.flavor" parameter set to "v2" and regressing out "percent.mt". The datasets were merged using the identified 3000 common features. The PCA was performed and batch effect between datasets during integration removed using harmony. Harmonised data were further reduced using UMAP on the 40 first dimensions of the corrected PCA projections. Clustering was performed using Louvain algorithm at the resolution 0.6. The different cluster cell types were labeled using a list of canonical markers[25] and SingleR package using the BlueprintEncodeData as a reference. Tumor fibroblasts were similarly subclustered with the resolution parameter set to 1.

### Cell-cell communication analysis

scRNA-Seq expression values and cell type cluster informations were used with the R package LIANA to obtain the consensus ranking of the implemented methods (using the "liana_wrap()" and "liana_aggregate()" functions with the default settings). The predicted interactions based on the consensus ranks were further represented as circos plots and heatmaps using the R packages circlize and ComplexHeatmap, respectively.

### Dataset analysis

The Cancer Genome Atlas (TCGA) colorectal cancer (TCGA-COAD and TCGA-READ) STAR - counts for protein coding genes of primary tumor samples (data release version 36.0) were extracted from the GDC portal (gdc.cancer.gov/) using the GenomicDataCommons R package and normalized using DESeq2 (median of ratio normalization or MRN). FFPE or FFPE validation samples (according to the biospecimen aliquot informations) were discarded ensuring that each patient was linked to a single sample. The final dataset was composed of unique expression values for 624 CRC patients. The CMS classification provided by Guinney et al.[57] was used. Gene signature scores were calculated as the mean of scaled $\log_2$ transformed expression values. The stromal score was calculated using the R estimate package. The immune landscape heatmap was generated using the ComplexHeatmap R package and the immune cell infiltration scores provided by Thorsson et al.[58] were used. To estimate the proportion of infiltrating lymphocytes, macrophages and M2 macrophages the provided CIBERSORTx relative immune abundance scores were multiplied by the leukocyte fractions. The Th17 score was calculated as the mean of scaled log2 gene expression values of the genes composing the signature (*IL17A*, *IL17F*, *IFNG*, *CD3E*, *CD4*, *STAT3*, *STAT5A*, *STAT1*, *STAT4*, *STAT6*, *AHR*, *RORC*, *RORA* and *TBX21*). The T-cell proliferation score was calculated using a published T-cell proliferation module gene set[59]. The iCAF and IL1R1+ iCAF gene sets defined in this manuscript (Supplementary Table 2) were similarly used to calculate iCAF and IL1R1+ iCAF scores.

The overall survival of CRC patients was analyzed using the GSE39582 dataset. The dataset was downloaded from the GEO portal using the GEOquery R package and the CMS classification published by Guinney et al.[57] was used. The Cox proportional hazard-ratio model as implemented in the survival R package was used to evaluate differences in overall survivals. Kaplan-Meier curves were rendered using ggplot2 and the ggkm R packages.

Gene set enrichment analyses (GSEA) were performed using the "GSEA" function implemented in the clusterProfiler R package. Preranked list of genes out the differential gene expression analyses were used with gene sets defined in this study (i.e. IL-1β, IL1R1+ iCAFs) or the hallmark gene sets downloaded using the R msigdbr package.

The Calon and Nishida dataset expression values for epithelial and stromal cells were downloaded from the GEO portal using the GEOquery R package.

### Mouse scRNA-Seq dataset

Raw data for the mouse colon GSE134225 dataset was downloaded from the GEO portal and processed using the R package Seurat (v4.3.0). The R package clustree (0.5.0) was used to visually explore the best suited clustering resolution (resolution = 0.4). The detected clusters were labelled using canonical markers for main cell types which were previously published[60].

### Tissue microarrays, immunohistochemistry, and immunofluorescence

Tissue microarrays were prepared for 106 primary CRC samples and all stainings were carried out according to previously published procedures[53]. Immunohistochemical staining was performed using an automated Benchmark XT device (Roche) using the Cell Conditioning 2 antigen retrieval solution. Antibodies against IL-1β (ab2105, abcam), αSMA (19245, Cell Signaling Technology) and FAP (ab53066, abcam) were used.

7 μm thick cryosections of subcutaneous MC38 tumors were prepared on a cryotome (CM1850 UV Cryostat, Leica). Sections were fixed in acetone and stained with DAPI and anti-PD-L1 APC antibody. Images were acquired with an Olympus IX83 microscope. The PD-L1 positive area was determined by applying a threshold in ImageJ.

Encapsulated tumor organoids and CAFs were incubated overnight with anti-NFκB/p65 RelA (8242, Cell Signaling Technology) overnight at 4 °C followed by the anti-rabbit Alexa 488 coupled secondary antibody. Then, directly labeled Alexa 647 coupled anti-Vimentin (ab195878, abcam) and Alexa 555 coupled anti-EPCAM (5488, Cell Signaling Technology) were incubated overnight at 4 °C and DNA stained with DAPI. Microphotographs were acquired, CAFs cells and nuclei outlined and the p65 nuclear and cytoplasmic staining intensities measured using Fiji.

CRC tissue sample sections from 5 donors were stained with anti-PDPN, anti-IL1R1 antibodies and DAPI. Images were acquired and the staining colocalization analyzed using Fiji and the "Colocalization_Finder" plugin.

### Reporting summary

Further information on research design is available in the Nature Portfolio Reporting Summary linked to this article.

## Data availability

The scRNA-Seq and RNA-Seq data generated in this study have been deposited at the European Genome-phenome Archive (EGA) under the accession number EGAS00001007205. The sequencing data generated in this study is of human origin. Hence, according to the European Data Protection Regulation, the data are available under restricted access, access can be obtained by contacting the corresponding author at Elisabeth.letellier@uni.lu. The procedure to access data is described in https://ega-archive.org/access/data-access. The scRNA-Seq publicly available data used in this study are available in the Gene Expression Omnibus (GEO) database under accession codes GSE81861, GSE146771, GSE132465, GSE144735, GSE134255. The microarray and RNA-Seq data used in this study are available in the Gene Expression Omnibus (GEO) database under accession codes GSE39582, GSE198697, GSE35602 and GSE39397. The TCGA-COAD and TCGA-READ RNA-Seq publicly available STAR counts gene expression data were obtained from the GDC portal [https://portal.gdc.cancer.gov/] and the corresponding Immune cell infiltration scores were obtained from the Supplemental Information released with the Thorsson et al.[58] research article [https://doi.org/10.1016/j.immuni.2018.03.023]. The Guinney CMS classifications for the GSE39582 and TCGA datasets were

obtained from the Synapse link provided by the authors [https://www.synapse.org/#!Synapse:syn2623706]. The Cancer hallmark gene sets were obtained from the Molecular Signatures Database [https://www.gsea-msigdb.org/gsea/msigdb/human/genesets.jsp?collection=H]. Source data are provided with this paper. The remaining data are available within the Article, Supplementary Information or Source Data file. Source data are provided with this paper.

## Code availability

The code used in this study has been deposited at https://gitlab.lcsb.uni.lu/mdm/koncina_et_al_2023.

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

## Acknowledgements

We thank the patients who kindly donated their samples and made this study possible. We thank all the contributing surgeons and nurses from the Centre Hospitalier Emile Mayrisch, the Laboratoire National de Santé and the Clinical and Epidemiological Investigation Centre of the LIH for their work with the patients. We would like to thank the Fondation Cancer for its support during the setup of the Luxembourgish CRC patient cohort. We also thank Prof. Dr. Michel Mittelbronn and the pathologists and macroscopy team from NCP/LNS. The authors would also like to thank their collaborators at the IBBL, Dr. Fay Betsou, Dr. Nikolai Goncharenko, Dr. Christelle Bahlawane and Amélie Gaignaux for the overall setup of the patient sample collection and the management of the cohort. We would also like to thank Dr. Lynn Bonetti for providing us with the Th17 gene signature. We would also like to thank the whole team of the animal facility at the University of Luxembourg, especially our veterinarian Jennifer Behm and our facility manager Djalil Coowar. Finally we would like to thank Professor Olivier de Wever for the kind gift of the Ct5.3 fibroblasts. Parts of the data processing presented in this manuscript were carried out using the HPC facility of the University of Luxembourg[61] (https://hpc.uni.lu). Figures 4a, 7 and Supplementary Fig. 4a were created with BioRender.com (https://biorender.com). The diagrams shown in Figs. 5b, 6a and Supplementary Fig. 1b were generated using Inkscape (https://inkscape.org/) and the mouse clipart used in Figs. 5b and 6a are adapted from "Mouse" from Servier Medical Art by Servier, licensed under a Creative Commons Attribution 3.0 Unported License (https://smart.servier.com)'. This work was supported by the Luxembourg National Research Fund [CORE/C16/BM/11282028 (L.E.), CORE/C20/BM/14591557 (L.E.), PoC/18/12554295 (L.E.), PRIDE Doctoral Research in the scope of the Doctoral Teaching Unit—CANBIO (PRIDE15/10675146/CANBIO) to N.M. and B.R. and MICROH PRIDE17/11823097 to T.M., as well as by the Fondation du Pélican de Mie and Pierre Hippert-Faber under the aegis of the Fondation de Luxembourg (E-AGR-0023-10-Z; T.M.), the Fondation Schumacher (B.R.), and the Fondation Gustave et Simone Prévot (L.E.). The project was further supported by CCC research grant (*Initiative Krebsforschung)* of the MedUni Vienna (D.H.), European Commission, SECRET ITN 859962 (D.H.) and Gesellschaft für Forschungsförderung Niederösterreich m.b.H.; LSC18-017 (D.H.)]. The collection of colorectal cancer samples is supported by the Fondation Cancer (L.E.) and the Doctoral School in Science and Engineering (T.M.) and the Department of Life Sciences and Medicine at the University of Luxembourg. The funders had no role in study design, data collection and analysis, decision to publish, or preparation of the manuscript.

## Author contributions

Conceptualization, Ko.E., N.M., P.V.I., D.H. and L.E.; Methodology, Ko.E., N.M., P.V.I., G.C., B.R., T.M., S.S., G.A., W.C., A.V.S., G.K., K.J., D.H., L.E.; Clinical Methodology, G.J., A.V., K.Y.E., K.L., Z.N. Formal analysis, Ko.E., N.M., P.V.I., G.C., B.R., G.A., T.M.; Investigation, Ko.E., N.M., P.V.I., G.C., G.A., T.M., B.R., W.C., G.K., R.F., S.M., U.P., Kl.E., H.R., S.A., H.M., H.S., M.J., Writing—original draft, N.M., E.Ko., and L.E.; Writing—review and editing, E.Ko., T.M., D.H., L.E. Funding acquisition, D.H. and L.E. Supervision L.E.

## Competing interests

The authors declare no competing interests.
