## [Peer Review File · Nature Communications]

IL1R1+ cancer-associated fibroblasts drive tumor development and immunosuppression in colorectal cancerREVIEWER COMMENTS

Reviewer #1 (Remarks to the Author): with expertise in CAFs, scRNAseq

In this study the authors present a comprehensive analysis of single cell RNA seq datasets of CRC patients and identify an inflammatory, IL1R1+ cancer associated fibroblast (CAF) implicated in tumor growth and immunosuppression. The authors use in vitro assays and genetic mouse models to demonstrate the role of IL1R1 CAFs in promoting tumor growth and invasion, as well as interaction with immune cells such as macrophages and T cells. While this study has significant translational relevance, it is not entirely novel as inflammatory CAFs have been studied and characterized across various stroma rich cancers such as PDAC, breast cancer, including CRC. I have a few reservations which if addressed would deem the publication of this study.

Comments

1. Please include cell numbers for each dataset analysed in the UMAP embedding in Fig 1A. Also, include a distribution plot of fibroblasts from each dataset (normal and CAFs) to understand how many fibroblasts are being analyzed.
2. It would be comprehensive to integrate all single cell datasets in the main figure to demonstrate that the IL1R1 CAF is shared across datasets and patients from different labs, ie integrate CLZ, Lee and Qian datasets. In the methods section, please describe in detail how the harmony integration was performed. For example, specify the variables used for "batch correction", diversity clustering penalty parameter, number of harmony dimensions.
3. In ED Figure 2B mCAFs do not express PDPN, instead they express RGS5 along with ACTA2 suggesting that these might be pericytes and have been erroneously annotated as CAFs. I recommend the authors to investigate this further and provide a dot plot with fibroblast markers from various published studies such as Dominguez & Muller et al., Cancer Discovery 2020 amongst others to support their current annotation.
4. Can the authors comment on how the fibroblast compartment remodels upon specific ablation of IL1R1 fibroblasts? Do myofibroblasts or the other cy-CAF1 fibroblasts take over in the absence of IL1R1 fibroblasts? Can these results be recapitulated in an orthotopic context?

Reviewer #2 (Remarks to the Author): with expertise in CAFs, CRC

The manuscript by Koncina and colleagues describes a novel cancer-associated fibroblast (CAF) population characterized by IL-1 receptor expression and prominent IL-1beta driven signaling. The authors identify that this CAF population can be assigned to a specific IL-1 beta rich signature, termed cyCAF-2. Indeed, in vitro assays display an autocrine IL-1 beta activation loop that promotes tumor growth in 3D assays. Furthermore, IL-1 beta activation triggered cyCAF-2 cells display immunosuppressive capacities by modulating macrophages, as well as T cell expansion. Moreover, the authors used a mouse model describing the tumor promoting effect of IL-1 receptor expressing fibroblasts. Finally, cyCAF-2 presence correlates with decreased survival in CMS4 patients. The article is well presented, conclusive and provides novel insights in the biology of CAFs. I therefore recommend for publication at Nature Communication, once my concerns are addressed. My major concern is the choice of the mouse model. Why did the authors choose the colVI mouse model? Is colVI expression prominent in cyCAF-2? Or do colVI expressing fibroblasts display elevated IL-1 receptor expression? Another issue is the choice of subcutaneous injection of the tumor cells, as the microenvironment is distinct to the one in the colon. An orthotopic model would be more suitable.

Other points, questions the authors should address:

- Why the datasets from Li and Zhang were chosen for the initial analysis? Do they have specific features in common with the inhouse data set, e.g. stage or subtype?
- The authors should comment on the IL1R1-expression of endothelial cells.
- Figure 1 consists mostly of transcriptomic data. The authors should try to validate some findings on

the protein level, e.g. by immunohistochemistry using the inhouse TMA.

- The authors show that CMS4 tumors have a cyCAF-2 signature. Does the inhouse TMA contain mostly CMS4 tumors?
- Fig. 2I shows apparently staining only on one patient sample. The authors should extend the number. As Figures 2I and J focus on podoplanin, Fig. 2H should include a staining for podoplanin as well.
- Taking the phosphorylation status in extended Figure 3E as readout, the activation status upon IL-1 beta treatment seems to be higher in NF compared to CAF. The authors should comment on this.
- Figure legend 3G should include details on how the respective analysis was performed. An example for CAFs in tumor proximity would be helpful.
- Do the employed mouse models resemble human CMS4 tumors?
- Do extended Figure 6C and Figure 6D refer to the same experiment? Do tumor bearing colVI-cre+ mice die at later time points?
- If I understand Figure 6L correctly, also cold tumors have a distinct TH17 score?

Reviewer #3 (Remarks to the Author): with expertise in CAFs, CRC, IL1

The manuscript by Koncina and colleagues addresses the pro-tumorigenic role of IL-1R+ CAFs in CRC. Based on single cell analyses the authors convincingly demonstrate the presence of these cells as a subset in CMS4 tumors and provide compelling functional in vivo data employing both pharmacological and genetic tools. The valuable conclusions are well supported by the data. Overall, the ms is well written and the experiments are performed at a high standard and I only have a few points that should be considered.

Specific points

1. The authors speculate that IL-1r+ CAFs mainly impact macrophage M2 -polarization and Th17 polarization of T cells. However, it would be nice if the authors could demonstrate this in more detailed manner and address the responsible factors in a more comprehensive and functional manner. It would be nice if conditioned supernatants or co culture experiments could be used for blocking/neutralization experiments. Moreover, the effects on tumor cells should be analyzed in more detail and for example supernatants from IL-1rKO CAFs compared to WT CAFs in terms of pro-mitogenic action on tumor cells.
2. CMS4 tumors are in general characterized by a strong TGFbeta signature which in principle would counteract formation of IL-1r+ CAFs. This should be experimentally tested and then the authors should try to address the question how these two apparently diverse subsets can be found both in CMS4 tumors and how/why one or the other cell type occur preferentially.

Reviewer #4 (Remarks to the Author): with expertise in cancer immunology

Koncina et al., presented a highly interesting manuscript about the role IL1R+ fibroblasts in colorectal cancer. The manuscript is well-written and especially the in vitro and in silico part is thoroughly executed using different datasets and experiments. Parts of the results section were very data and information-dense, making it a little challenging to follow. Some additional points:

- In figure 1D the authors show IL1R1 expression on the CAFs and NF of 1 patient. Can the authors provide some more information of this specific patient (age, gender, tumor stage etc). Does the expression of IL1R1 expression vary between patients? Does addition of IL1b induce/enhance expression of IL1R1? Are the CAFs of the patient cultured for many passages (that does change the phenotype), does the level change during passaging?
- Fig 2J; Similar to my previous remark, please provide patient info regarding CAF-5, CAF-6 and CAF-7. Why did the authors only show data on the expression of PDPN? What about aSMA, or even more interesting IL1R1? This would nicely show variance between patients and how they respond to IL1b.
- In Fig3A,B only the technical replicates are shown. Please provide the 3 independent experiments in

1 figure (as is already nicely done in 2J, 4B, 4C, 5C, 5E) and use a nested ANOVA (or present the other experiments in the supplement so readers can appreciate the biological variance between experiments). Similar holds true for fig3D, fig4d

- Fig 3G: it would be helpful if the authors would show a representative microscopy image for this figure, not only the quantified data. I also miss a description of this assay in the M&M. If I understand it correctly this experiment is performed with CAFs from 1 patient. Is there difference when using CAFs from different patients (f.e. CAFs that have lower IL1R1 vs higher IL1R1?)

- I miss a description about the use of ligand-receptor analyses framework LIANA in the M&M

- Figure 6: the authors only show data for the MC38 until day 19 (fig6D) or day 22 (ext fig6C). To fully appreciate the effect of IL1R1+CAF's in tumor growth, the authors also should show data of later time points. How many mice were used for extfig6C and has it been repeated to confirm these data?

In figure 6 the authors mainly focus on Th17, while in their previous (in vitro) experiments they showed an effect on (M2) macrophages. Can the authors confirm their in vitro data on macrophages in vivo?

- Fig6D,F, I, F, K: To appreciate the biological variance between experiments, please also provide the other (not-shown) independent experiments in the supplement.

- Fig6E, J: why do the authors present a normalized tumor volume? Please provide the actual tumor volume of the mice.

- The title suggests the authors investigated the role of IL1R1+ CAFs on tumor development, while they rather looked into tumor progression

We would like to thank all the Reviewers for their kind comments highlighting the interesting nature of our manuscript and are very much thankful for their recommendations on how to improve it. We believe that our updated version now addresses most of the concerns raised by the reviewers and integrates new findings, which certainly improve the quality of the manuscript.

Before addressing the reviewer's comments, we would like to draw attention to the fact that **we decided to rename cyCAFs to iCAFs and mCAFs to myCAFs**. Indeed, we are further showing here the overlap with previously described signatures (see the response to R1 point 3 below). We believe that **this decision will further help to harmonize and avoid adding any additional confusion** within the CAF field, as it was recently highlighted in the review published by the Tuveson team ([10.1016/j.ccell.2023.02.015](https://doi.org/10.1016/j.ccell.2023.02.015)).

Reviewer #1: with expertise in CAFs, scRNAseq

In this study the authors present a comprehensive analysis of single cell RNA seq datasets of CRC patients and identify an inflammatory, IL1R1+ cancer associated fibroblast (CAF) implicated in tumor growth and immunosuppression. The authors use in vitro assays and genetic mouse models to demonstrate the role of IL1R1 CAFs in promoting tumor growth and invasion, as well as interaction with immune cells such as macrophages and T cells. While this study has significant translational relevance, it is not entirely novel as inflammatory CAFs have been studied and characterized across various stroma rich cancers such as PDAC, breast cancer, including CRC. I have a few reservations which if addressed would deem the publication of this study.

Comments

1. Please include cell numbers for each dataset analysed in the UMAP embedding in Fig 1A. Also, include a distribution plot of fibroblasts from each dataset (normal and CAFs) to understand how many fibroblasts are being analyzed.

We now include a stacked bar chart showing the number of cells analyzed in each dataset and tissue type. Additionally, **we provide the number of NFs and CAFs in the 3 different datasets** in Extended Fig. 1A (table on the right panel and see below).

New Extended Figure 1A

Proportion of cells in normal and tumor tissue in each dataset (left barchart) and number of total cells as well as normal and tumor fibroblasts identified in each of the 3 scRNA-Seq datasets (right table).

Please refer to the new panel A of Extended Fig. 1.

Besides, we would like to reply to the comment about the novelty of our study. Indeed, some publications already highlighted IL-1 β signaling in fibroblasts (10.3892/ol.2019.10784, 10.1016/j.yexcr.2011.05.023, 10.3390/ijms22094960). However, to our knowledge, most studies relied on cell lines in an IL-1 β treatment context. Indeed, these studies not only focus on another cancer type, but additionally describe the tumor cells as releasing IL-1 β (10.3892/ol.2019.10784 and 10.1016/j.yexcr.2011.05.023 in an oral cancer context). This is in stark contrast to our study where **we show compelling evidence that in CRC, IL-1 β can be released by myeloid cells and CAFs themselves**, the latter fueling an autocrine signaling loop, **but not by tumor cells**.

- It would be comprehensive to integrate all single cell datasets in the main figure to demonstrate that the IL1R1 CAF is shared across datasets and patients from different labs, ie integrate CLZ, Lee and Qian datasets. In the methods section, please describe in detail how the harmony integration was performed. For example, specify the variables used for “batch correction”, diversity clustering penalty parameter, number of harmony dimensions.

We thank the reviewer for the comment. We have now integrated the different scRNA cohorts composing the 3 datasets (excepting the Li cohort from CLZ as the cell number contribution and quality were lower than for the remaining cohorts). When sub clustering the tumor fibroblast compartment, **we validate the identification of an IL1R1 enriched cluster in the integrated dataset** (see new Extended Fig. 2C-E) which includes the cells which we labeled IL1R1⁺ iCAF in our independent dataset analysis (Extended Fig. 2A-B). In addition, **we show that this cluster is not composed of a single cohort but that the IL1R1⁺ iCAF population is nicely present and distributed in all the datasets** (see Extended Fig. 2D and Fig. r1-2 for reviewing purposes). Furthermore, **we now describe more exhaustively how the harmony integration**

was performed, including the variables being used for the batch correction and the number of dimensions retained to perform the harmony integration.

Panels added to extended figure 2 to show the IL1R1⁺ CAF cluster in the integrated CZ-Lee-Qian dataset **A.** UMAP plot showing the main cell types identified in the integrated (Cole, Zhang, Lee SMC, Lee KUL3 and Qian) dataset (left) and the distribution of normal and tumor tissue (middle) as well as the distribution of the cohorts composing the dataset (right). **B.** UMAP plot showing the identified CAF clusters (labelled 0-10) in the integrated dataset (left), the identified iCAF, IL1R1⁺ iCAF and myCAF group of clusters (middle) and the distribution of the cohorts composing each cluster (right). **C.** Violin plot showing the *IL1R1* expression and piecharts showing the percentage of *IL1R1* expressing cells in each CAF cluster.

The integration of the different datasets has been added to Extended Fig. 2C-E as well as the distribution plot (Extended Fig. 2D, right panel).

- In ED Figure 2B mCAFs do not express PDPN, instead they express RGS5 along with ACTA2 suggesting that these might be pericytes and have been erroneously annotated as CAFs. I recommend the authors to investigate this further and provide a dot plot with fibroblast markers from various published studies such as Dominguez & Muller et al., Cancer Discovery 2020 amongst others to support their current annotation.

We acknowledge the difficulty to rule out that RGS5⁺ and ACTA2⁺ cells we labeled as being myCAFs might in fact be pericytes. Dominguez et al. (10.1158/2159-8290.CD-19-0644) pointed out that cells which were previously identified as being CAFs by Puram et al. in head and neck cancer might in fact be pericytes (10.1016/j.cell.2017.10.044). While it is difficult, if not currently impossible to conclude, it is important to note that CAFs might also originate from pericytes. Such a plasticity has for instance been described by Hosaka et al. (10.1073/pnas.1608384113) where PDGF-BB treated pericytes adopt a fibroblast signature over time. Furthermore, in line with such a plasticity, ACTA2⁺ CAFs have been reported to majorly derive from pericytes in mouse colon tumors (10.1053/j.gastro.2021.11.037).

We provide here further analyses supporting that our labeling is in line with the labeling proposed by others. To this end **we cross compared our myCAF labeling to the original labels proposed by the authors of the datasets**. For instance, while the original Lee et al. labels indeed identified a subset of the myCAF cluster as being pericytes, it should be emphasized that a **very recent study which reused the same cohort (SMC + KUL3) is in line with our labeling as they attributed these cells to the fibroblast compartment**. It

should also be noted that the labeling of Lee et al. is a transposition of labels from Kinchen et al. (10.1016/j.cell.2018.08.067), which were generated in an IBD context and thus ignoring the CAF phenotype. **We were also able to confirm that Qian et al. labeled the same cells as being fibroblasts.** Finally, **we also looked at the pan-CAF atlas dataset** for the expression of ACTA2, RGS5 and MCAM (10.1038/s41467-022-34395-2). As shown in Fig. r1-3 below, **clusters attributed to the fibroblast compartment are positive for these markers.** To further highlight that we are aware of this difficulty of segregating pericytes from myCAFs in scRNA sequencing data, we now included a comment in the discussion.

Figure r1-3 for reviewing purposes

A. Barchart showing the cell type labels provided by the original authors of the SMC, KUL3 (Lee) and Qian cohorts.
B. Bubble heatmap showing the expression of RGS5, MCAM and ACTA2 in fibroblast clusters identified in the single cell CAF atlas (GSE210347; EndoMT CAF = endothelial-to-mesenchymal transition CAF, pnCAF = peripheral nerve-like CAF, apCAF = antigen-presenting CAF, NF = normal fibroblast).

We now include a comment in the results on lines 123-125 to address the difficulty to discriminate myCAFs and pericytes in scRNA-Seq datasets.

- Can the authors comment on how the fibroblast compartment remodels upon specific ablation of IL1R1 fibroblasts? Do myofibroblasts or the other cy-CAF1 fibroblasts take over in the absence of IL1R1 fibroblasts? Can these results be recapitulated in an orthotopic context?

We thank the reviewer for this comment and suggestion to investigate how the fibroblast compartment remodels upon *IL1R1* deletion. We isolated fibroblasts from *IL1R1* deficient (Cre^+) and control (Cre^-) mice to analyze the expression of selected fibroblast markers by flow cytometry. As shown in the new Extended Fig. 8H, we can observe that when *IL1R1* is absent, fibroblasts seem to adopt a more myofibroblastic phenotype characterized by lowered expression of FAP and PDPN and upregulated expression of α SMA, PDGFR α and PDGFR β .

New Extended Figure 8H

Expression of CAF markers measured by flow-cytometry on colon fibroblasts isolated from *IL1R1* deficient (Cre^+) and control (Cre^-) mice.

We now include this data in Extended Fig. 8H.

Reviewer #2: with expertise in CAFs, CRC

The manuscript by Koncina and colleagues describes a novel cancer-associated fibroblast (CAF) population characterized by IL-1 receptor expression and prominent IL-1 β driven signaling. The authors identify that this CAF population can be assigned to a specific IL-1 β rich signature, termed cyCAF-2. Indeed, in vitro assays display an autocrine IL-1 β activation loop that promotes tumor growth in 3D assays. Furthermore, IL-1 β activation triggered cyCAF-2 cells display immunosuppressive capacities by modulating macrophages, as well as T cell expansion. Moreover, the authors used a mouse model describing the tumor promoting effect of IL-1 receptor expressing fibroblasts. Finally, cyCAF-2 presence correlates with decreased survival in CMS4 patients. The article is well presented, conclusive and provides novel insights in the biology of CAFs. I therefore recommend for publication at Nature Communication, once my concerns are addressed.

1. My major concern is the choice of the mouse model. Why did the authors choose the colVI mouse model? Is colVI expression prominent in cyCAF-2? Or do colVI expressing fibroblasts display elevated IL-1 receptor expression?

We thank the reviewer for the positive feedback on our study and appreciate the insightful comments. Nowadays different promoters have been described to target the fibroblast population with *Col6a1* and *Col1a1* being the most popular ones. We chose *Col6a1* as it has been linked to highly efficient recombination in the gut ([10.1038/s41586-020-2166-3](https://doi.org/10.1038/s41586-020-2166-3)). Our aim was to **knock-out the expression of *Il1r1* specifically in the complete fibroblast compartment encompassing the IL1R1⁺ iCAF population**. As shown in Extended Fig. 8A,C and Fig. r2-1 for reviewing purposes, while *Il1r1* is highly expressed in mouse fibroblasts (and to a lesser extent in endothelial cells), the expression of both, *Col6a1* and *Col1a1* are very specific to fibroblasts. Nevertheless, we acknowledge that more innovative models, such as the Split-Cre system, might allow to more specifically target a given gene in a subpopulation of fibroblasts in the future.

2. Another issue is the choice of subcutaneous injection of the tumor cells, as the microenvironment is distinct to the one in the colon. An orthotopic model would be more suitable.

We thank the reviewer for this comment. Whereas we fully agree that an orthotopic injection model would be physiologically more relevant and better represent the tumor microenvironment than a subcutaneous one, we would like to emphasize that **we already used two different and complementary *in vivo* approaches**: a conditional KO model (see Fig. 6A-B,D-E) and a pharmacological model (Fig. 6B,I-J), to validate our results obtained *in silico* and *in vitro* in the human context. We are currently establishing the orthotopic model (via colonoscopy-based injection) in our group. However, to date, we are still optimizing the methodology and most importantly waiting for ethical approval to achieve this project. Taking the time required for the ethical approval and reliable establishment of the model into consideration, would significantly delay the project to address this specific point. Additionally, we do not believe that adding an orthotopic model would fundamentally change the major take-home messages of the paper, which are backed-up by the *in silico* and *in vitro* work we performed. **We believe that the combination of the different approaches we used in the study strengthens by itself the main message of the paper** *i.e.* the characterization of a clinical relevant IL1R1 positive CAF subtype. Nevertheless, establishing this model in our laboratory is one of our priorities to use it in upcoming CAF projects.

3. Other points, questions the authors should address:

- a. Why the datasets from Li and Zhang were chosen for the initial analysis? Do they have specific features in common with the inhouse data set, e.g. stage or subtype?

There is no specific reason, except that **when we started our study, the Li and Zhang datasets were the only ones publicly available**. We did not select for a specific subtype nor a CRC stage in the scRNA-Seq analysis as this might have significantly limited the analysis. As shown in comment 2 to R1, **we have now integrated all scRNA-Seq datasets and were able to identify the same IL1R1 subtype in most of the datasets**, further validating that we identify the *IL1R1*⁺ CAF population cluster in different datasets.

Please refer to the new Extended Fig. 2C-E.

- b. The authors should comment on the IL1R1-expression of endothelial cells.

We appreciate the suggestion and agree that we need to better discuss the expression of IL1R1 in endothelial cells. We observed that in addition to fibroblasts, **endothelial cells also express higher levels of *IL1R1***, suggesting that the phenotype we observe might be partially driven by endothelial cells expressing *IL1R1*. However, as shown in the heatmap and UMAP plot of Extended Fig. 8A and 8C, respectively, as well as in the Fig. r2-3b below, **endothelial cells minimally express *Col6a1***, therefore **we can rule out the possibility that they contributed to the *in vivo* phenotype we observed in the conditional KO mouse model**. Nevertheless, deciphering the relevance of IL1R1 signaling in endothelial cells in the CRC context needs to be considered in the future.

Figure r2-3b for reviewing purpose

A. UMAP plot showing the expression of *IL1R1*, *COL6A1* and *COL1A1* in fibroblasts and endothelial cells in the human integrated scRNA-Seq dataset (CLZ + Lee + Qian). **B.** UMAP plot showing the expression of *Il1r1*, *Col6a1* and *Col1a1* in fibroblasts and endothelial cells in the mouse scRNA-Seq dataset (GSE134255).

We now added a sentence in the results section addressing the expression of *IL1R1* in endothelial cells (see lines 279-282).

- c. Figure 1 consists mostly of transcriptomic data. The authors should try to validate some findings on the protein level, e.g. by immunohistochemistry using the inhouse TMA.

We thank the reviewer for the comment. We have now performed additional immunofluorescence stainings on sections of primary tumor samples to show the presence of *IL1R1*⁺ fibroblasts, which coexpress PDPN. Unfortunately, in our hands, the anti-PDPN antibody we tested on our in-house TMA didn't work out (Invitrogen MA5-16267) as we failed to observe a specific immunostaining for PDPN (see Fig. r2-3c for reviewing purposes below). We nevertheless **added a colocalization quantification on the**

additional immunofluorescence stainings we performed and show the overlap between IL1R1 and PDPN on an alternative representative immunohistochemistry microphotograph. We updated Fig. 2I accordingly to include our new analysis. **We also show the expression of IL1R1 on CAFs from different patients**, which varies according to the patient (Extended Fig. 1H).

Figure r2-3c for reviewing purposes

Attempt to stain PDPN on our in-house TMA using the antibody from Invitrogen (MA5-16267). Scale bar = 50 μ m.

Updated Figure 2I

IL1R1 and PDPN co-localization in immunofluorescence stainings of a human tumor samples (left and upper right; Scale bar = 50 μ m) and beeswarm plot showing the normalized Mander's colocalization coefficient of IL1R1 and PDPN on sections measured on sections from 5 different patients (lower right, distinct patients are encoded as different dot colors).

New Extended Figure 1H.

IL1R1 expression measured by flow cytometry in CAF cell lines isolated from three distinct patients.

Panel in Fig. 2I has been updated and panel in Extended Fig. 1H has been added.

- d. The authors show that CMS4 tumors have a cyCAF-2 signature. Does the inhouse TMA contain mostly CMS4 tumors?

We didn't select the patients of our cohort based on the CMS status. **We now provide in the updated Supplementary table 2 the summary of the CMS subtyping of our CRC cohort** (subtyped using the R package CMSCaller and the RNA-Seq counts of SOCS tumor samples). **Accordingly, 29.5% of the TMA tumor samples are CMS4**, whereas 15.9% are CMS1, 25.0% CMS2 and 12.5% CMS3 (17.0% didn't show a clear signature specific to one of the four CMS subtypes). Using the same cohort we now further analyzed how IL-1 β correlates with FAP and α SMA in the different subtypes. Interestingly and along with our previous results, **the correlation between IL-1 β and FAP is particularly high in CMS1 and CMS4 compared to the other subtypes.**

variable	n
Age	
≤ 65	19
> 65	87
Gender	
female	32
male	74
Stage	
1	15
2	42
3	37
4	10
unknown	2
Tumor localisation	
proximal colon	36
distal colon	40
rectosigmoid	7
rectum	21
unknown	2
CMS	
CMS1	14 (15.9%)
CMS2	22 (25.0%)
CMS3	11 (12.5%)
CMS4	26 (29.5%)
NOLBL	15 (17.0%)
Not subtyped	18

Updated Supplementary table 2

Descriptive statistics including the CMS subtyping of our in-house CRC cohort (106 patients analysed in the TMA in Fig. 2)

New Extended Figure 3D

Correlation between FAP⁺ and IL-1β⁺ staining (upper panel) and αSMA⁺ and IL-1β⁺ staining (lower panel) identified after IHC staining on tissue microarray sections of our established in-house CRC cohort and split by CMS (n=73 patients with identified CMS out of the total of 106 available TMAs).

Please refer to Supplementary table 2 and Supp Data 3D.

- e. Fig. 2I shows apparently staining only on one patient sample. The authors should extend the number. As Figures 2I and J focus on podoplanin, Fig. 2H should include a staining for podoplanin as well.

We performed additional immunofluorescence stainings of tumor tissue samples for PDPN and IL1R1. As already mentioned in point 3c raised by reviewer #2, we were not able to setup a specific immunostaining for PDPN (using Invitrogen MA5-16267) on FFPE slides and therefore on our in-house TMA (see Fig. r2-3c for reviewing purposes). Nevertheless, **we further completed the analysis by showing that PDPN and IL1R1 colocalize** by showing a colocalization quantification (by measuring the Mander's colocalization coefficient, ranging from 0 to 1) of IL1R1 with PDPN on immunofluorescence sections from **five different patients** and updated the representative images shown in the manuscript.

Updated Figure 2I

IL1R1 and PDPN co-localization in immunofluorescence stainings of a human tumor sample (left and upper right; Scale bar = 50 μ m) and beeswarm plot showing the normalized Mander's colocalization coefficient of IL1R1 and PDPN measured on sections from 5 different patients (lower right, distinct patients are encoded as different dot colors).

We have included more patients and updated Fig. 2I accordingly. We have added a quantification analysis of the co-localization between PDPN and IL1R1.

- f. Taking the phosphorylation status in Extended Fig. 3E as readout, the activation status upon IL-1 beta treatment seems to be higher in NF compared to CAF. The authors should comment on this.

We thank the reviewer for the comment. Our impression was that the phosphorylation status between NF and CAF on the blot was not different and that the slight difference was rather due to the unequal loading control levels. To rule out such a difference, we performed WB analyses on additional NF and CAF pairs. Although the phosphorylation status is variable within NF and CAF pairs, CAFs tend to exhibit higher phosphorylation levels (Fig. r2-3f for reviewing purposes). Furthermore, using the RNA-Seq dataset GSE198697, we confirm that NF κ B target genes are upregulated in CAFs when compared to NFs as shown by the heatmap in the new Extended Fig. 1I.

Figure r2-3f for reviewing purposes

p65, phosphorylated p65 and β-actin expression in NFs (P175, P177) and CAFs (P177 and CT5.3) upon IL-1β stimulation (0.1 and 1 ng/ml), as assessed by western blotting).

We have now included this data in Fig. 11

- g. Figure legend 3G should include details on how the respective analysis was performed. An example for CAFs in tumor proximity would be helpful.

We repeated the experiment shown in Fig. 3G using CAFs from a different donor (updated Fig. 3G showing CAF-8 cells and Extended Fig. 4H showing CAF-7 cells). In addition, we now show representative immunohistochemistry stainings for EPCAM, VIM and p65 RelA as well as outlines for CAFs being proximal and distal to tumor spheroids as being used for p65 N/C quantification (New Extended Fig. 4F).

We have now updated Fig. 3G, added the Extended Figures 4F and 4H as well as updated the Material and Methods section to better describe how the analyses shown in Fig. 3G and Extended Fig. 4H were performed.

h. Do the employed mouse models resemble human CMS4 tumors?

We used the MC38 mouse CRC cell line for our in vivo model. It should be noted that CMS4 is a feature that is not inherent to tumor cells. Indeed, the classification of colorectal cancer (CRC) into CMS is based on bulk sequencing data of human colorectal cancer samples, which include tumor cells, tumor stromal cells such as fibroblasts as well as immune cells. The CMS4 subtype of CRC, which is known as the mesenchymal phenotype, is based on a substantial infiltration of stromal cells in these tumors; the epithelial tumor cells in these subsets are always of CMS type 2 or 3. This was recently further emphasized in studies that compared single cell sequencing and bulk transcriptomic from the same patients (see Joanito et al. [10.1038/s41588-022-01100-4](https://doi.org/10.1038/s41588-022-01100-4)). Additionally, we now stained resected MC38 tumors and could indeed observe α SMA, PDGFR α and FAP positive cells, clearly showing that those tumors are infiltrated by fibroblasts which further validates the model used in the present study (see new Extended Fig. 8E).

New Extended Figure 8E

Presence of fibroblasts in MC38 tumors shown by α SMA, FAP and PDGFR α immunofluorescence stainings as well as DAPI stained DNA content. Scale bar = 50 μ m.

We have now added the Extended Fig. 8E to support the validity of the MC38 injection model.

i. Do Extended Fig. 6C and Fig. 6D refer to the same experiment? Do tumor bearing colVI-cre+ mice die at later time points?

Extended Fig. 6C (moved to Extended Fig. 8F in the updated version of the manuscript) and Fig. 6D indeed refer to the same experiment. The Kaplan-Meier plot shows the time to reach humane endpoint. As the Kaplan-Meier curve suggests, and supported by the significant Mantel-Cox test, *IL1R1* deficient (Cre⁺)

mice die at a later timepoint than the control (Cre⁻) mice (no *IL1R1* deficient mouse reached humane endpoint during the maximal timeframe of our experiment).

j. If I understand Figure 6L correctly, also cold tumors have a distinct TH17 score?

Fig. 6L shows the Th17 score in *IL1R1*^{hi} and *IL1R1*^{lo} patients without further considering the immune infiltration or immune cell activity that would define cold and hot tumors. We hope that this explanation clarifies the results in the manuscript.

Reviewer #3: with expertise in CAFs, CRC, IL1

The manuscript by Koncina and colleagues addresses the pro-tumorigenic role of IL-1R+ CAFs in CRC. Based on single cell analyses the authors convincingly demonstrate the presence of these cells as a subset in CMS4 tumors and provide compelling functional in vivo data employing both pharmacological and genetic tools. The valuable conclusions are well supported by the data. Overall, the ms is well written and the experiments are performed at a high standard and I only have a few points that should be considered. Specific points

1. The authors speculate that IL-1r+ CAFs mainly impact macrophage M2 -polarization and Th17 polarization of T cells. However, it would be nice if the authors could demonstrate this in more detailed manner and address the responsible factors in a more comprehensive and functional manner. It would be nice if conditioned supernatants or co culture experiments could be used for blocking/neutralization experiments. Moreover, the effects on tumor cells should be analyzed in more detail and for example supernatants from IL-1rKO CAFs compared to WT CAFS in terms of pro-mitogenic action on tumor cells.

We thank the reviewer for this very valuable comment. We have now further analyzed the CAF compartment in order to identify potential cytokines that might be involved in M2 polarization. To this end, we followed a methodology like the one described by Kobayashi et al 2022 ([10.1053/j.gastro.2021.11.037](https://doi.org/10.1053/j.gastro.2021.11.037)). The method consists in identifying the overlap of genes upregulated in *IL1R1*⁺ iCAFs (union of the *IL1R1*⁺ iCAF signature genes identified in the 3 scRNA-Seq datasets) with a gene set of secreted cytokines known to exert macrophage/monocytes chemotaxis. While overlaying both gene sets, **we identified 3 potential candidates: CCL2, CXCL12 and IL6**. We then looked whether IL-1 β can increase the expression of the potential hit candidates. While *CXCL12* was not increased after IL-1 β simulation, ***CCL2* and *IL6* were both up-regulated following treatment with IL-1 β** (Fig. 5F-G, new Extended Fig. 7A-B). Strikingly, *CCL2* has long been described to shape macrophage polarization and fibroblast derived *CCL2* might thereby recruit monocytes to the tumor bed and induce their polarization towards M2 macrophages. Indeed, we have previously shown that CRC CAFs, in coculture with tumor cells and monocytes / macrophages, substantially induce *CCL2* in both CAFs and macrophages – a process dependent on CAF derived M-CSF expression – and thus generate an enhanced monocyte recruiting microenvironment ([10.1016/j.canlet.2021.07.006](https://doi.org/10.1016/j.canlet.2021.07.006)). Of note, IL-6, a bona fide NF κ B and IL-1 β target gene

([10.1074/jbc.M707692200](https://doi.org/10.1074/jbc.M707692200)), is involved in the upregulation of CCL2 in monocytes ([10.1182/blood.V91.1.258](https://doi.org/10.1182/blood.V91.1.258)). Interestingly, *IL6* is induced in CAFs upon co-culture with patient matched tumor organoids to a greater extent than in NFs (see new Extended Fig. 7C) and plays also a crucial role in augmenting the M2 polarization of macrophages ([10.1371/journal.pone.0094188](https://doi.org/10.1371/journal.pone.0094188)).

New Extended Figure 7A-C

A-B. Violin plot showing the expression of *CXCL12* (**A**), and *IL6* (**B**) upon stimulation of fibroblasts with IL-1 β . **C.** Heatmap showing the expression of identified hit candidates and NF- κ B target genes in NFs and CAFs upon coculture with tumor organoids (GSE198697).

Concerning Th17 cells, we show in the manuscript the involvement of IL1R1 signaling in the differentiation of Th17 cells (Fig. 6F,K), which is in line with previous reports ([10.4049/jimmunol.1300387](https://doi.org/10.4049/jimmunol.1300387) and [10.1016/j.immuni.2009.02.007](https://doi.org/10.1016/j.immuni.2009.02.007)).

We now added the new analysis as Fig. 5F-H and Extended Fig. 7A-C.

2. CMS4 tumors are in general characterized by a strong TGFbeta signature which in principle would counteract formation of IL-1r+ CAFs. This should be experimentally tested and then the authors should try to address the question how these two apparently diverse subsets can be found both in CMS4 tumors and how/why one or the other cell type occur preferentially.

We agree with R2 that IL-1 β and TGF- β signaling are closely linked. We have now performed experiments to investigate the crosstalk between both signaling pathways. In agreement with the literature, **we observed that TGF- β induces a myofibroblastic phenotype** (characterized by increased expression of α SMA), which could be **partially reverted by the induction of the IL-1 β signaling pathway** (new Extended Fig. 4I). Interestingly, we observed that neither IL-1 β nor TGF- β alone could induce the expression of FAP. However, when we simultaneously triggered both signaling pathways, we observed a large increase of FAP expression (new Extended Fig. 4I). This finding is quite intriguing and highlights the complex interaction between both pathways, which cannot be reduced to a negative feedback interaction alone. Moreover, there is increasing evidence for the simultaneous presence of different types of CAFs at spatially distinct sites in the TME ([10.1053/j.gastro.2021.11.037](https://doi.org/10.1053/j.gastro.2021.11.037)), which – in line with the scRNA-Seq dataset analyses which identified different CAF subsets – strongly suggest that topological distinct areas might experience different signals and thus show different CAF phenotypes. However, further studies are needed to understand why these 2 (or more) distinct subsets are found in CMS4 tumors.

We added the new analysis as Extended Fig. 4I and completed the results on lines 188-200.

New figure 4I

CAF phenotype induced by IL-1 β and TGF- β activation crosstalk. CAFs (primary cultures of CAF-5, CAF-6 and CAF-7) were treated with either IL-1 β (5 ng/ml), TGF- β (5 ng/ml) or both cytokines together and the expression of PDGFR β , FAP, α SMA and PDPN measured by flow cytometry. MFI values obtained on the three different CAFs were normalized (non-centered scaling). Different data point shapes show technical replicates for the three different CAFs. Tukey post-hoc test following a nested ANOVA design (*/*♣/♦: $p < 0.001$; * vs. untreated control, ♣ vs. IL-1 β treated and ♦ vs. TGF- β treated).

Reviewer #4: with expertise in cancer immunology

Koncina et al., presented a highly interesting manuscript about the role IL1R+ fibroblasts in colorectal cancer. The manuscript is well-written and especially the *in vitro* and *in silico* part is thoroughly executed using different datasets and experiments. Parts of the results section were very data and information-dense, making it a little challenging to follow. Some additional points:

1. In figure 1D the authors show IL1R1 expression on the CAFs and NF of 1 patient. Can the authors provide some more information of this specific patient (age, gender, tumor stage etc). Does the expression of IL1R1 expression vary between patients? Does addition of IL1b induce/enhance expression of IL1R1? Are the CAFs of the patient cultured for many passages (that does change the phenotype), does the level change during passaging?

We thank the reviewer for the valuable comment. We now provide the patient information in the figure legend of panel 1D. We further assessed the expression of IL1R1 by FACS on additional patients. As anticipated from the RNA-Seq analysis shown in Extended Fig. 4B, we observe differences in IL1R1 expression among the different patients (new Extended Fig. 1H). Additionally, we also determined the IL-1 β -induced expression of IL1R1, however it seems that the induction of IL1R1 by IL-1 β is also patient dependent. As suggested by the reviewer, culturing CAFs *in vitro* is challenging and affected by multiple parameters. As these cells are removed from their original environment, their profile might change once plated in a plastic dish and in particular other passages. It should be noted that, so far, the development

of reliable *in vitro* models which preserve the CAF phenotype has been neglected. There is an urgent need in the scientific community working in the CAF field to invest time into defining robust *in vitro* models to perform mechanistic studies on CAFs. As we have indeed noticed gradual changes in the fibroblast's phenotypes upon prolonged cultivation (after passage 10-12), including the expression of IL1R1 (data not shown), we only use CAFs and NFs at low passages (between p2 and p10).

2. Fig 2J; Similar to my previous remark, please provide patient info regarding CAF-5, CAF-6 and CAF-7. Why did the authors only show data on the expression of PDPN? What about α SMA, or even more interesting IL1R1? This would nicely show variance between patients and how they respond to IL1b.

We thank the reviewer for this comment and have added more information on the patient-derived CAFs in the Material and Methods section. We now completed the analysis shown in Fig. 2J using additional CAF markers and show them in Extended Fig. 4I (where CAFs are stimulated with both IL-1 β and or TGF- β). When looking in our RNA-Seq dataset we confirm that *PDPN* expression is increased upon stimulation with IL-1 β (see Fig. r4-2 for reviewing purposes below). Interestingly, only two out of the four treated CAFs also showed an upregulation of *IL1R1* (Fig. r4-2). This might be linked to patient-dependent differential basal expression levels of IL1R1 as already suggested by our transcriptomic data presented in Extended Fig. 4B and the additional FACS quantification of IL1R1 shown in the new Extended Fig. 1H. A second reason for the differential response of IL1R1 expression upon IL1B stimulation might be the timing of analysis after stimulation induction. We will explore the time course of IL1R1 expression upon stimulation with different growth factors, cytokines or chemokines in more depth in future experiments.

Figure r4-2 for reviewing purposes

Expression of *IL1R1* and *PDPN* in IL-1β (1 ng/ml) treated CAFs (CT5.3, P20, P32 and P42; RNA-Seq data).

New figure 4I

CAF phenotype induced by IL-1β and TGF-β activation crosstalk. CAFs (primary cultures of CAF-5, CAF-6 and CAF-7) were treated with either IL-1β (5 ng/ml), TGF-β (5 ng/ml) or both cytokines together and the expression of PDGFRβ, FAP, αSMA and PDPN measured by flow cytometry. MFI values obtained on the three different CAFs were normalized (non-centered scaling). Different data point shapes show technical replicates for the three different CAFs. Tukey post-hoc test following a nested ANOVA design (*/*/*/*: p < 0.001; * vs. untreated control, ♣ vs. IL-1β treated and ♦ vs. TGF-β treated).

Please refer to the new data in Extended Fig. 1H and Extended Fig. 4I.

- In Fig3A,B only the technical replicates are shown. Please provide the 3 independent experiments in 1 figure (as is already nicely done in 2J, 4B, 4C, 5C, 5E) and use a nested ANOVA (or present the other experiments in the supplement so readers can appreciate the biological variance between experiments). Similar holds true for fig3D, fig4d

We thank the reviewer for the comment as we indeed have been able to show for most of the data multiple NF and CAF pairs. We would like to underline that **NFs and CAFs are not kept indefinitely in culture** as they rapidly undergo senescence after few passages. For this reason, it is sometimes difficult, if not impossible, to repeat experiments using the very same matched NF and CAF pairs. Nevertheless, as pointed out by the reviewer, **we now repeated the above-mentioned experimental setting to provide additional data** and used a nested Anova analysis as we already did in other figures shown in the manuscript.

4. Fig 3G: it would be helpful if the authors would show a representative microscopy image for this figure, not only the quantified data. I also miss a description of this assay in the M&M. If I understand it correctly this experiment is performed with CAFs from 1 patient. Is there difference when using CAFs from different patients (f.e. CAFs that have lower IL1R1 vs higher IL1R1?)

We now provide a representative image in Extended Fig. 4F to show the staining and illustrate how we performed the quantification of p65 (see pink dotted lines delineating the nuclei and cells which were quantified). In addition, as highlighted in the response to reviewer 3, we observe that NFκB target genes are up-regulated upon coculture with tumor organoid (Extended Fig. 4G).

CAF outline and nucleus
CAF proximal to tumor spheroid
CAF distant to tumor spheroid

New Extended Figure 4F

Representative microphotograph showing the Vimentin (in red) and p65 (in green) staining on CAF – LS174 cocultures (scale bar = 50 μm). The dotted lines show nuclei which were measured for p65 staining intensity.

5. I miss a description about the use of ligand-receptor analyses framework LIANA in the M&M

We thank the reviewer for highlighting this lacking information. We now completed the Material and Method section to include the LIANA analysis.

See Material and Method section on lines 547 to 551.

6. Figure 6: the authors only show data for the MC38 until day 19 (fig6D) or day 22 (ext fig6C). To fully appreciate the effect of IL1R1+CAFs in tumor growth, the authors also should show data of later time points. How many mice were used for extfig6C and has it been repeated to confirm these data? In figure 6 the authors mainly focus on Th17, while in their previous (in vitro) experiments they showed an effect on (M2) macrophages. Can the authors confirm their in vitro data on macrophages in vivo?

We already mentioned in the discussion of the original manuscript that we were not able to validate the M2 polarization phenotype, which we observed in vitro, in our mouse CRC model. This could be due to multiple reasons including the chosen time point to analyze the immune cell composition in tumors. It might well be that the experimental endpoint might be too late to characterize such a phenotype while characterizing macrophages at an earlier time point might confirm the phenotype which we were able to observe *in vitro*. Another possibility is that the TME in the subcutaneous setting is not supporting all our data collected on CRC CAFs. Undoubtedly, and as already outlined in response to reviewer #2, an orthotopic injection model would be physiologically more relevant and better represent the tumor microenvironment than a subcutaneous one. We are currently establishing the orthotopic model (via colonoscopy-based injection) in our lab. Most importantly, we don't have the ethical approval in place to achieve this experiment. Taking the time required for the ethical approval and reliable establishment of the model into consideration, would significantly delay the project to address this specific point.

7. Fig6D,F, I, F, K: To appreciate the biological variance between experiments, please also provide the other (not-shown) independent experiments in the supplement.

We have added individual experiments in updated Fig. 6J,K and Extended Fig 8G. Please also refer to the comment below.

8. Fig6E, J: why do the authors present a normalized tumor volume? Please provide the actual tumor volume of the mice.

We decided to show a joint representation of the tumor volumes from multiple experiments in adequation with the statistical analysis we performed as, despite our efforts, we observed a tumor growth batch effect between experiments (Fig. 6E). **To allow the appreciations of the biological variance between experiments, we now added the independent measurements for Fig. 6E as absolute volumes in Extended Fig. 8G and show absolute volumes in the updated Fig. 6J (Anakinra).**

9. The title suggests the authors investigated the role of IL1R1+ CAFs on tumor development, while they rather looked into tumor progression

Thank you very much for the comment. We agree that our *in vivo* model reflects tumor progression. However, we would like to underscore that our study also examines a critical signaling pathway that has the potential to transform normal fibroblasts into CAFs, an integral aspect of the initial phases of tumor growth. Moreover, the term development refers to the entire process of the formation and growth of a tumor, which includes initiation and progression, thus we would like to kindly request that the original title be retained.

2 Additional analyses

Below is a summary of some additional analyses trying to address some of the highlighted points.

2.1 Fibroblasts by scRNA-Seq dataset

2.2 TMA by CMS (Fig. 2H)

(a) Correlation of FAP with IL1B

(b) Correlation of αSMA with IL1B

Figure 1: SOCS TMA by CMS (reponse to #R2-3d, see Section 1.2)

Table 1: Distribution of tumour samples analysed in the TMA by CMS.

cms	n
CMS1	14 (13.2%)

cms	n
CMS2	22 (20.8%)
CMS3	11 (10.4%)
CMS4	26 (24.5%)
NOLBL	15 (14.2%)
NA	18 (17.0%)

Guo, Wei, Cuiyu Zhang, Xia Wang, Dandan Dou, Dawei Chen, and Jingxin Li. 2022. "Resolving the difference between left-sided and right-sided colorectal cancer by single-cell sequencing." *JCI insight* 7 (1): e152616. <https://doi.org/10.1172/jci.insight.152616>.

Joanito, Ignasius, Pratyaksha Wirapati, Nancy Zhao, Zahid Nawaz, Grace Yeo, Fiona Lee, Christine L. P. Eng, et al. 2022. "Single-cell and bulk transcriptome sequencing identifies two epithelial tumor cell states and refines the consensus molecular classification of colorectal cancer." *Nature Genetics* 54 (7): 963–75. <https://doi.org/10.1038/s41588-022-01100-4>.

Kieffer, Yann, Claire Bonneau, Tatiana Popova, Roman Rouzier, Marc-Henri Stern, and Fatima Mechta-Grigoriou. 2020. "Clinical Interest of Combining Transcriptomic and Genomic Signatures in High-Grade Serous Ovarian Cancer." *Frontiers in Genetics* 11: 219. <https://doi.org/10.3389/fgene.2020.00219>.

Pelka, Karin, Matan Hofree, Jonathan H. Chen, Siranush Sarkizova, Joshua D. Pirl, Vjola Jorgji, Alborz Bejnood, et al. 2021. "Spatially Organized Multicellular Immune Hubs in Human Colorectal Cancer." *Cell* 184 (18): 4734–4752.e20. <https://doi.org/10.1016/j.cell.2021.08.003>.

REVIEWERS' COMMENTS

Reviewer #1 (Remarks to the Author):

The authors have satisfactorily addressed my queries. I have no further questions.

Reviewer #2 (Remarks to the Author):

My concerns have been well addressed by the authors.

Reviewer #3 (Remarks to the Author):

The authors have nicely addressed my concerns and the manuscript has significantly improved overall.

Reviewer #4 (Remarks to the Author):

The authors have addressed most of my points, I do however have some remarks/questions regarding the answers of the authors to some of my comments.

- Comment 1: I may have overlooked it, but I cannot find the patient information that should have been added in the legend of panel 1D. I also miss the patient characteristics from new extended figure 1H. This is valuable information.
- Comment 2: Similar to my previous remark, I cannot find any patient information (age, gender, tumor stage etc) of CAF-5, CAF-6 and CAF-7 as mentioned by the authors.
- Comment 6: Old ext fig6C (new ext fig 8F): please provide information on the number of mice used and mention whether this experiment has been repeated or not.
- Comment 7/8: The new figure 6J is exactly the same as the old figure 6J only the labeling of the axes changed from % of mean ctrl to cm³, didn't it change when showing the absolute volumes instead of the normalized values?

Response to reviewers

We would like to thank all the Reviewers for acknowledging that we now addressed most of the previous recommendations and were able to improve the manuscript with their helpful comments. We also thank R4 for the additional comments concerning some precisions which were still missing in the manuscript.

Reviewer #4: with expertise in cancer immunology

The authors have addressed most of my points, I do however have some remarks/questions regarding the answers of the authors to some of my comments.

1. Comment 1: I may have overlooked it, but I cannot find the patient information that should have been added in the legend of panel 1D. I also miss the patient characteristics from new extended figure 1H. This is valuable information.
2. Comment 2: Similar to my previous remark, I cannot find any patient information (age, gender, tumor stage etc) of CAF-5, CAF-6 and CAF-7 as mentioned by the authors.

We thank the reviewer for highlighting this still lacking information. We now **provided these details in the new supplementary table 3** and point to it within the corresponding figure legends.

3. Comment 6: Old ext fig6C (new ext fig 8F): please provide information on the number of mice used and mention whether this experiment has been repeated or not.

The mouse survival data we are showing in supplementary figure 8f include two independent experiments with 7 ColVI^{Cre-} *IL1R1^{fl/fl}* and 6 ColVI^{Cre+} *IL1R1^{fl/fl}* mice. The third independent experiment, which showed similar effect and is included in the tumor volume graph in Fig 6e, was terminated on day 19 post injection, and samples were used for the assessment of the PD-L1 expression. This has now been clarified in the Figure legends.

4. Comment 7/8: The new figure 6J is exactly the same as the old figure 6J only the labeling of the axes changed from % of mean ctrl to cm³, didn't it change when showing the absolute volumes instead of the normalized values?

Indeed, we previously presented the Anakinra *in-vivo* data (figure 6J) as scaled volumes to be in line with the transgenic mice study (in which scaling by experiment was necessary and able to overcome the interexperimental variability we were facing). As we applied a global scaling to the Anakinra treatment data, the relative differences remain indeed the same. Thus, the **current manuscript shows the scaled data only for the transgenic mice study (in the main figure 6e) alongside the tumor volumes (in supplementary figure 8g) and the tumor volumes for the Anakinra treatment experiment which indeed only affects the axis labeling as noticed by R4.**